# A Stochastic Optimization Framework for Private and Fair Learning From Decentralized Data

## Abstract

Machine learning models are often trained on sensitive data (e.g., medical records and race/gender) that is distributed across different "silos" (e.g., hospitals). These *federated learning* models may then be used to make consequential decisions, such as allocating healthcare resources. Two key challenges emerge in this setting: (i) maintaining the *privacy* of each person's data, even if other silos or an adversary with access to the central server tries to infer this data; (ii) ensuring that decisions are *fair* to different demographic groups (e.g., race/gender). In this paper, we develop a novel algorithm for private and fair federated learning (FL). Our algorithm satisfies *inter-silo record-level differential privacy* (ISRL-DP), a strong notion of private FL requiring that each silo's communicated messages satisfy record-level differential privacy. In addition to being differentially private, our framework can be used to promote different fairness notions, including demographic parity and equalized odds. We prove that our algorithm converges under mild smoothness assumptions on the loss function (even in nonconvex settings), whereas prior work required strong convexity for convergence. As a byproduct of our analysis, we obtain the first convergent algorithm for ISRL-DP optimization of nonconvex-strongly concave min-max loss functions in federated learning. This convergent DP optimization algorithm is a valuable contribution in its own right. Additionally, our experiments demonstrate the state-of-the-art fairness-accuracy tradeoffs of our algorithm across different privacy levels. Compared to existing state of the art, we obtained an average of around 64% reduction in demographic parity fairness violation and 95% lower for equalized odds.

## 1 Introduction

Many important decisions are being assisted by machine learning (ML) models (e.g., loan approval or criminal sentencing). Without intervention, ML modes may discriminate against certain demographic groups (e.g., race, gender). For instance, Amazon developed a ML-based recruiting software that showed a strong bias against hiring women for technical jobs (Dastin, 2018). *Algorithmic fairness* research aims to develop algorithms that promote equitable treatment of different demographic groups by correcting biases that may lead to unfair outcomes.

Despite the development of numerous fair learning algorithms, two key challenges impede their real-world application: (1) Training fair models requires access to *sensitive data* (e.g., age, race, gender) in order to ensure fairness of predictions with respect to these attributes. However, data protection and privacy regulations (like E.U.'s General Data Protection Regulation and California's Consumer Privacy Act) restrict the usage of sensitive demographic consumer data (GDP, 2016; BUKATY, 2019). (2) Training data is often *distributed* across different organizations, such as hospitals or banks, who may not share their data with third parties.

To address obstacle (1), prior works (Jagielski et al., 2019; Mozannar et al., 2020; Tran et al., 2021; 2022; Lowy et al., 2023) have used *differential privacy* (Dwork et al., 2006) to preserve the privacy of the sensitive data during fair model training. Informally, differential privacy (DP) ensures that no adversary can infer much more about any individual piece of sensitive data than they could have inferred had that piece of data

never been used. While these works address the first challenge, they fail to address the second challenge, since they require centralized access to the full data.

In this work, we address the two aforementioned challenges via fair private *federated learning* (McMahan et al., 2017) under an appropriate notion of differential privacy. Federated learning (FL) is a distributed learning framework in which silos collaborate to train a global model by exchanging focused updates, often with the orchestration of a central server. By permitting silos to collaborate without sharing their sensitive local data, FL offers an ideal solution to challenge (2).

Although FL offers some privacy benefits to silos via local storage of data, this is not sufficient to prevent sensitive data from being leaked: model parameters or updates can leak data, e.g. via gradient or model inversion attacks (Li et al., 2024b;a). To prevent sensitive data from being leaked during FL, we will require the full transcript of silo $i$'s sent messages (e.g., local gradient updates) to be differentially private. This privacy requirement is known as *inter-silo record-level differential privacy* (ISRL-DP) (Lowy & Razaviyayn, 2023; Liu et al., 2022), defined formally in Section 2. For example, if the silos are hospitals, then ISRL-DP preserves the privacy of each patient's record, even if an adversary with server access colludes with the other hospitals to try to decode the data of hospital $i$.

**Prior work.** There are several existing work on centralized private and fair learning (Jagielski et al., 2019; Lowy et al., 2023; Tran et al., 2021), on private federated learning (Lowy et al., 2022b; Lowy & Razaviyayn, 2021; Girgis et al., 2021; Gao et al., 2024), and on fair FL (Ezzeldin et al., 2023). However, *the literature on private and fair federated learning is sparse*. In fact, the only related works we are aware of are due to Rodríguez-Gálvez et al. (2021); Ling et al. (2024); Padala et al. (2021); Gu et al. (2022). The work of Rodríguez-Gálvez et al. (2021) does not prove any ISRL-DP guarantee for their algorithm, nor do they provide a convergence guarantee. The work of Ling et al. (2024) only guarantees convergence for *strongly convex* loss functions, limiting its applicability in a wide range of modern ML models. For example, linear/logistic regression and deep learning loss functions are not strongly convex, limiting the applicability of their developments. Moreover, the algorithm of Ling et al. (2024) promotes a particular form of fairness notion known as *balanced performance fairness,* which is less popular than other notions such as *demographic parity* or *equalized odds.* The work of Padala et al. (2021) goes through a two-stage training. They first train a fair model using FairSGD, in a non-private manner, and then train a DP model which emulates the output of the FairSGD model. But that final trained model is not necessarily fair. On the other hand, the work of Gu et al. (2022) cannot work with mini-batches of data due to their method of enforcing fairness, limiting its use in applications with large training datasets.

**Contributions.** Motivated by the shortcomings of prior works, our work addresses the following question:

> Can we develop an algorithm for fair and private federated learning that provably converges, even with loss functions that are not necessarily strongly convex?

To answer this question, we extend a framework from Lowy et al. (2023) for promoting fairness and ISRL-DP with respect to sensitive attributes in a federated learning setting. Our framework is flexible, covering different fairness notions such as demographic parity and equalized odds. Further, our algorithm provides:

1. **Guaranteed ISRL-DP and convergence:** We prove that our ISRL-DP algorithm converges for any smooth (*potentially non-convex*) loss function, even when mini-batches of data are used (i.e. stochastic optimization). Thus, our algorithm can be used in large-scale FL settings, where full batch training is not feasible.

2. **State-of-the-art empirical performance**: our ISRL-DP algorithm achieves significantly improved fairness-accuracy tradeoffs on benchmark tasks across different privacy levels. For example, the equalized odds *fairness violation of our algorithm is* $95\%$ *lower than the previous state-of-the-art* (Ling et al., 2024) for the same fixed accuracy level. Additionally, our algorithm even outperforms a set of significant centralized DP fair baselines that do not provide the strong protection of ISRL-DP Tran et al. (2021).

FIX
[9rJo]

Furthermore, our analysis yields a significant theoretical byproduct: a convergent FL algorithm for ISRL-DP optimization in nonconvex-strongly concave min-max optimization. This convergent DP optimization algorithm is a valuable contribution in its own right. Moreover, our framework extends to *hybrid centralization* settings, where some data features are centralized and others are decentralized (e.g., centralization of sensitive and decentralization of non-sensitive data). We discuss this in a detailed manner in section 4.

## 2    Problem Setting and Preliminaries

Consider a federated learning setting with $N$ silos (e.g., hospitals or banks), each of which has data that is partitioned into sensitive and non-sensitive data divisions: $\{Z_j = (X_j, Y_j), S_j\}_{j=1}^N$, where $(X_j, Y_j) = \{x_{j,i}, y_{j,i}\}_{i=1}^{\tilde{n}}$, $S_j = \{s_{j,i}\}_{i=1}^{\tilde{n}}$, and $\tilde{n} = n/N \in \mathbb{N}$ is the number of local samples per silo. $x_{j,i} \in \mathcal{X}$ are the non-sensitive features, $s_{j,i} \in [k] \triangleq \{1, \ldots, k\}$ are the discrete sensitive attributes (e.g. race, gender), and $y_{j,i} \in [l] \triangleq \{1, \ldots, l\}$ are the ground-truth labels.[1] Let $\hat{y}_\theta(x)$ denote the model predictions parameterized by $\theta$, and $\ell(\theta, x, y) = \ell(\hat{y}_\theta(x), y)$ be a loss function (e.g. cross-entropy loss). Our goal is to (approximately) solve the empirical risk minimization (ERM) problem

$$\min_\theta \left\{ \widehat{\mathcal{L}}(\theta) := \frac{1}{N\tilde{n}} \sum_{j=1}^N \sum_{i=1}^{\tilde{n}} \ell(\theta, x_{ji}, y_{ji}) \right\} \tag{1}$$

in a fair manner, while maintaining the differential privacy of the sensitive data $\{S_j\}_{j=1}^n$ under ISRL-DP. We consider two different notions of fairness in this work:[2]

FIX [qZng]

**Definition 1** (Fairness Notions). Let $\mathcal{A} : \mathcal{X} \to \mathcal{Y}$ be a classifier.

- $\mathcal{A}$ satisfies *demographic parity* (Dwork et al., 2012) if the predictions $\mathcal{A}(X)$ are statistically independent of the sensitive attributes.

- $\mathcal{A}$ satisfies *equalized odds* (Hardt et al., 2016a) if the predictions $\mathcal{A}(X)$ are conditionally independent of the sensitive attributes given $Y = y$ for any $y \in \mathcal{Y}$.

The choice of fairness notion depends on the application at hand (See Chouldechova & Roth (2020) for discussion.)

It has been demonstrated that achieving perfect fairness is *impossible* for a differentially private algorithm that also achieves non-trivial accuracy (Cummings et al., 2019). Therefore, we focus on developing an algorithm that minimizes a certain measure of *fairness violation* on the given dataset $Z$. Fairness violations can be quantified in various ways; see, Dwork et al. (2012); Hardt et al. (2016a); Lowy et al. (2022a) for an overview. As an example, if demographic parity is the desired fairness criterion, we can quantify the (empirical) demographic parity violation using the following measure:

$$\max_{\hat{y} \in \mathcal{Y}} \max_{s \in \mathcal{S}} \left| \hat{p}_{\hat{Y}|S}(\hat{y}|s) - \hat{p}_{\hat{Y}}(\hat{y}) \right|, \tag{2}$$

where $\hat{p}$ represents the empirical probability computed from the data $(Z, \{\hat{y}_i\}_{i=1}^n)$. Note that the demographic parity violation in equation 2 is zero if and only if demographic parity is satisfied. The expression for Equalized Odds violation for the binary case is given in Appendix E.1.

NEW [qZng]

Next, we define differential privacy. Following the DP fair learning literature (Jagielski et al., 2019) and motivated by the discussion in the Introduction, we consider a relaxation of DP, in which only the *sensitive attributes* require privacy.[3] In the centralized setting, we say $Z$ and $Z'$ are *adjacent with respect to sensitive data* if $Z = \{(x_i, y_i, s_i)\}_{i=1}^n$, $Z' = \{(x_i, y_i, s_i')\}_{i=1}^n$, and there is a unique $i \in [n]$ such that $s_i \neq s_i'$.

---

[1]Our algorithm and analysis readily extends to the case in which silo data sets contain different numbers of samples, via standard techniques (see e.g., Lowy & Razaviyayn (2023))

[2]Our method can also handle any other fairness notion that can be defined in terms of statistical (conditional) independence, such as equal opportunity. However, our method cannot handle all fairness notions: for example, false discovery rate and calibration error are not covered by our framework.

[3]However, the convergence guarantee of our algorithm easily extends to the case where privacy of the entire data set is needed.

**Definition 2** (Differential Privacy w.r.t. Sensitive Attributes)**.** Let $\varepsilon \geqslant 0$, $\delta \in [0, 1)$. A randomized algorithm $\mathcal{A}$ is $(\varepsilon, \delta)$-*differentially private (DP) w.r.t. sensitive attributes $S$* if for all pairs of data sets $Z, Z'$ that are *adjacent w.r.t. sensitive attributes*, we have

$$\mathbb{P}(\mathcal{A}(Z) \in O) \leqslant e^{\varepsilon}\mathbb{P}(\mathcal{A}(Z') \in O) + \delta, \tag{3}$$

for all measurable sets $O \subseteq \mathcal{Y}$.

In the context of FL with $N$ silos, we say two distributed datasets $Z = (Z_1, \ldots, Z_N)$ and $Z' = (Z'_1, \ldots, Z'_N)$ with $Z_j = \{(x_{j,i}, y_{j,i}, s_{j,i})\}_{i=1}^{\tilde{n}}$ $Z'_j = \{(x_{j,i}, y_{j,i}, s'_{j,i})\}_{i=1}^{\tilde{n}}$ are adjacent if for every $j \in [N]$, there is one $i \in [\tilde{n}]$ such that $s_{j,i} \neq s'_{j,i}$ given that the datasets for all other silos are fixed.

FIX [ZGVZ]

**Definition 3** (Inter-Silo Record-Level DP)**.** A federated learning algorithm $\mathcal{A}$ is $(\varepsilon, \delta)$-inter-silo-record-level DP (ISRL-DP) if, for each $j \in [N]$, the full transcript of silo $j$'s sent messages satisfies (3) for all adjacent distributed datasets $Z, Z'$ and any fixed settings of other silos' data.

Essentially ISRL-DP is a guarantee where, for each silo, the entire transcript of messages sent by that silo must be differentially private with respect to changing one individual's record in that silo, while the other silos' data are fixed. In contrast, central DP assumes a trusted curator/server that can access the pooled data and only requires the final released output to be private. Thus, ISRL-DP is stronger in the federated setting: even if the server or other silos inspect the messages sent by a target silo, they should not be able to infer much about any one individual in that silo. For more details about ISRL-DP, please refer to Lowy & Razaviyayn (2023).

NEW [ZGVZ]

**Scope of the privacy guarantee.** Our ISRL-DP guarantee is with respect to changes in the sensitive attributes $S$, while conditioning on the non-sensitive features and labels $(X, Y)$. This is useful in settings where the features and labels are available for model training, but demographic attributes are restricted or legally sensitive. For example, a hospital can use patients' demographic information to train a fair model without substantially revealing any individual patient's sensitive attribute to the server or other silos.

The guarantee protects against additional leakage from the algorithm's use of $S$ in the fairness regularizer, but it does not prevent inference of $S$ from $(X, Y)$ themselves when these variables are correlated. In settings where $(X, Y)$ are private or serve as strong proxies for $S$, one should instead use a stronger privacy model, such as full-record DP or an inferential privacy framework. By post-processing property of DP (Dwork & Roth, 2014), Definition 3 ensures that the model parameters and the messages broadcast by the central server and are also DP.

NEW [qZng, ZGVZ]

As discussed in Section 1, Definition 2 is useful if a company wants to train a fair model, but is unable to use the sensitive attributes collected in another silo (and is needed to train a fair model) due to privacy concerns and laws. Following Lowy et al. (2023), we shall impose the reasonably practical assumption that all data sets contain at least $\rho$-fraction of every sensitive attribute for some $\rho \in (0, 1)$.

## 3 Private and Fair Federated ERM Framework

A popular method in the literature to enforce fairness is to introduce a regularizer that penalizes the model for making unfair decisions (Zhang et al., 2018; Donini et al., 2018; Baharlouei et al., 2020). Let $S$, $Y$, and $\hat{Y}$ be the random variables corresponding to sensitive attributes, actual output, and predictions by the models. The regularization approach to fair ERM jointly optimizes for accuracy and fairness by solving

$$\min_{\theta} \left\{ \widehat{\mathcal{L}}(\theta) + \lambda \mathcal{D}(\widehat{Y}, S, Y) \right\},$$

where $\mathcal{D}$ is a measure of (conditional) statistical dependence (based on the fairness notion used) between the sensitive attributes $S$ and the predicted outputs $\hat{Y}$. The dependency of $\mathcal{D}$ on $S$, $\hat{Y}$, and/or $Y$ varies for different fairness notions. For instance, for demographic parity, $\mathcal{D}$ just depends on $S$ and $\hat{Y}$, while equalized odds $\mathcal{D}$ also depends on $Y$. The parameter $\lambda \geqslant 0$ controls the trade-off between accuracy and fairness. Inspired by the strong performance of Lowy et al. (2022a; 2023), we use variations of the $\chi^2$ divergence as our $\mathcal{D}$.

**Definition 4** ($\chi^2$ Divergence)**.** The $\chi^2$ Divergence between two probability mass functions $P(x)$ and $Q(x)$ over the support of $X$ is defined as

$$\chi^2(P||Q) = \sum_{x \in X} Q(x) \left( \frac{P(x)}{Q(x)} - 1 \right)^2$$

The choice of this divergence was motivated by the theoretical results and strong empirical performance of this divergence on stochastic fair optimization highlighted by the work of Lowy et al. (2022a). They provided extensive analysis on using this divergence using the following arguments:

- (Lowy et al., 2022a) proposed an unbiased estimator for the population $\chi^2$ divergence term, enabling the regularizer's use on mini-batches and ensuring convergence guarantees for stationarity.

- They demonstrated that the $\chi^2$ divergence between model outputs and sensitive labels serves as an upper bound for fairness violations, such as demographic parity and equalized odds. Hence, minimizing this divergence ensures the tightening of fairness violations.

FIX [qZng]

For demographic parity, we would ideally like to use $\mathcal{D}(\widehat{Y}, S) \triangleq \chi^2(p_{\widehat{Y}, S} || p_{\widehat{Y}} p_S)$ as our regularizer, where the *true joint distribution* for the random variables $\widehat{Y}$ and $S$ is given by $p_{\widehat{Y}, S}$ and marginals are given by $p_{\widehat{Y}}, p_S$, respectively. However, since the true distribution of $(\widehat{Y}, S)$ is unknown in practice, we resort to an empirical estimate of the regularizer: $\widehat{\mathcal{D}}(\widehat{Y}, S) \triangleq \chi^2(\hat{p}_{\widehat{Y}, S} || \hat{p}_{\widehat{Y}} \hat{p}_S)$, where the empirical joint distribution for the random variables $\widehat{Y}$ and $S$ is given by $\hat{p}_{\widehat{Y}, S}$ and marginals by $\hat{p}_{\widehat{Y}}$, $\hat{p}_S$ respectively. Similarly, for equalized odds, $\mathcal{D}'(\widehat{Y}, S, Y) \triangleq \chi^2(p_{\widehat{Y}, S|Y} || p_{\widehat{Y}|Y} p_{S|Y})$, and we use $\widehat{\mathcal{D}}(\widehat{Y}, S, Y) \triangleq \chi^2(\hat{p}_{\widehat{Y}, S|Y} || \hat{p}_{\widehat{Y}|Y} \hat{p}_{S|Y})$ in practice. We write the full expressions of these regularizers in Appendix A.

For concreteness, we consider demographic parity in what follows, but note that our developments extend easily to equalized oddds. Our approach to enforcing fairness is to augment (1) with the $\chi^2$ regularizer and privately solve:

$$\min_\theta \left\{ \text{FERMI}(\theta) := \widehat{\mathcal{L}}(\theta) + \lambda \widehat{\mathcal{D}}(\widehat{Y}_\theta(X), S) \right\}. \qquad \text{(FERMI obj.)}$$

The empirical divergence $\widehat{\mathcal{D}}$ is an asymptotically unbiased estimator of population divergence $\mathcal{D}$ (Lowy et al., 2022a), suggesting that solving (FERMI obj.) should generalize well to the corresponding population risk minimization problem.

The next question we address is: *how do we solve equation FERMI obj. in a distributed fashion, while satisfying ISRL-DP?* It is not obvious how to obtain statistically unbiased estimators of the gradients of $\widehat{\mathcal{D}}(\widehat{Y}_\theta(X), S)$ without directly computing $\nabla_\theta \widehat{\mathcal{D}}(\widehat{Y}_\theta(X), S)$ over the entire data set. But computing the gradient over the entire data set is not possible in the federated learning setting, since each silo stores its data locally in a decentralized manner.

Fortunately, Lowy et al. (2022a) gives us a statistically unbiased estimator through a min-max problem formulation. For feature input $x$, let the predicted class labels be given by $\hat{y}(x, \theta) = j \in [l]$ with probability $\mathcal{F}_j(x, \theta)$, where $\mathcal{F}(x, \theta) \in [0,1]^l$ is differentiable in $\theta$, and $\sum_{j=1}^l \mathcal{F}_j(x, \theta) = 1$. For instance, $\mathcal{F}(x, \theta) = (\mathcal{F}_1(x, \theta), \dots, \mathcal{F}_l(x, \theta))$ could represent the output of a neural net after softmax layer or the probability label assigned by a logistic regression model. Then we have the following min-max re-formulation of (FERMI obj.):

**Theorem 5** (Lowy et al. (2022a))**.** *There are differentiable functions $\widehat{\psi}_{ji}$ such that (FERMI obj.) is equivalent to*

$$\min_\theta \max_{W \in \mathbb{R}^{k \times l}} \left\{ \widehat{F}(\theta, W) := \widehat{\mathcal{L}}(\theta) + \lambda \frac{1}{N\tilde{n}} \sum_{j=1}^N \sum_{i=1}^{\tilde{n}} \widehat{\psi}_{ji}(\theta, W) \right\}. \qquad (4)$$

*Further, $\widehat{\psi}_{ji}(\theta, W)$ is strongly concave in $W$ for any $\theta$.*

---

**Algorithm 1** SteFFLe: Stochastic Private Fair Federated Learning

---
1: **Input**: $\{Z_j = \{x_{j,i}, y_{j,i}\}_{i=1}^{\tilde{n}}, \{s_{j,i}\}_{i=1}^{\tilde{n}}\}_{j=1}^{N}$, $\theta_0 \in \mathbb{R}^{d_\theta}$, $W_0 = 0 \in \mathbb{R}^{k \times l}$, step-sizes $(\eta_\theta, \eta_w)$, fairness parameter $\lambda \geqslant 0$, iteration number $T$, minibatch size $|B_t| = m \in [\tilde{n}]$, set $\mathcal{W} \subset \mathbb{R}^{k \times l}$, noise parameters $\{\sigma_{j,w}^2, \sigma_{j,\theta}^2\}_{j=1}^{N}$.
2: Compute $\widehat{P}_S^{-1/2} = \mathrm{diag}(\widehat{p}_S(1)^{-1/2}, \ldots, \widehat{p}_S(k)^{-1/2})$, where $\widehat{p}_S(r) := \frac{1}{N\tilde{n}} \sum_{j=1}^{N} \sum_{i=1}^{\tilde{n}} \mathbb{1}_{\{s_{ji}=r\}} \geqslant \rho > 0$.
3: **for** $t = 0, 1, \ldots, T-1$ **do**
4:    Central server sends $\theta_t, W_t$ to all silos.
5:    **for** $j \in [N]$ **in parallel do**
6:       Silo $j$ draws a mini-batch $B_t$ of data points $\{(x_{j,i}, y_{j,i}), s_{j,i}\}_{i \in B_t}$.
7:       Silo $j$'s non-sensitive division computes stochastic gradient $g_{t,j} := \frac{1}{|B_t|} \sum_{i \in B_t} \nabla_\theta \ell(x_{j,i}, y_{j,i}, \theta_t)$ where $\ell(x_{j,i}, y_{j,i}, \theta_t) = \mathrm{loss}(\mathcal{F}(x_{j,i}, \theta_t), y_{j,i})$ and $\mathcal{F}$ is the Neural Network used to compute the probabiliy distribution of the predicted label.
8:       Silo $j$ sends $\{\mathcal{F}(x_{j,i}, \theta_t), \mathrm{Clip}_{L_f}(\nabla \mathcal{F}(x_{j,i}, \theta_t)), j, i\}_{i \in B_t}$ to sensitive data division.
9:       Silo $j$'s sensitive division computes noisy sensitive stochastic gradients $h_{t,j,\theta} := \frac{1}{|B_t|} \sum_{i \in B_t} \nabla_\theta \widehat{\psi}_{j,i}(\theta_t, W_t) + u_{t,j}$ and $h_{t,j,w} := \frac{1}{|B_t|} \sum_{i \in B_t} \nabla_w \widehat{\psi}_{j,i}(\theta_t, W_t) + V_{t,j}$, where $u_{t,j} \sim \mathcal{N}(0, \sigma_\theta^2 \mathbf{I}_{d_\theta})$ and $V_{t,j}$ is a $k \times l$ matrix with independent random Gaussian entries $(V_{t,j})_{q,r} \sim \mathcal{N}(0, \sigma_w^2)$.
10:      Silo $j$ broadcasts $g_{t,j}$, $h_{t,j,\theta}$, and $h_{t,j,w}$ to the central server.
11:   **end for**
12:   Central server updates $\theta_{t+1} \leftarrow \theta_t - \frac{\eta_\theta}{N} \sum_{j=1}^{N} [g_{t,j} + \lambda h_{t,j,\theta}]$ and $W_{t+1} \leftarrow \Pi_\mathcal{W}\left(W_t + \frac{\lambda \eta_w}{N} \sum_{j=1}^{N} h_{t,j,w}\right)$ where $\Pi_\mathcal{W}(x) = \mathrm{argmin}_{y \in \mathcal{W}} ||x - y||_2$.
13: **end for**
14: Pick $\hat{t}$ uniformly at random from $\{1, \ldots, T\}$.
15: **Return:** $\hat{\theta}_T := \theta_{\hat{t}}$.

---

The functions $\widehat{\psi}_{ji}$ are given explicitly in Appendix B. With Theorem 5, we can now claim that: for any batch on a particular silo $\mathcal{B}_j$ with size $m \in [\tilde{n}]$, the gradients (with respect to $\theta$ and $W$) of $\frac{1}{Nm} \sum_{j=1}^{N} \sum_{i \in \mathcal{B}_j} \ell(x_{ji}, y_{ji}; \theta) + \lambda \widehat{\psi}_{ji}(\theta, W)$ are statistically unbiased estimators of the gradients of $\widehat{F}(\theta, W)$, if $\mathcal{B}$ is drawn uniformly from $Z$. However, when differential privacy of the sensitive attributes is also desired, the formulation (4) presents some challenges, due to the non-convexity of $\widehat{F}(\cdot, W)$. Lowy et al. (2023) solve this problem in the centralized setting, but the proposed method may leak central data to the server and does not satisfy ISRL-DP.

Next, we develop our distributed ISRL-DP fair learning algorithm.

## 3.1 ISRL-DP Fair Federated Learning via SteFFLe

Our algorithm for privately solving the min-max FL problem equation 4 is given in Algorithm 1. Algorithm 1 is essentially a noisy distributed variation of *stochastic gradient descent ascent* (SGDA). Gaussian noise is added to each silo's sensitive stochastic gradients $\nabla_\theta \widehat{\psi}, \nabla_w \widehat{\psi}$ to ensure ISRL-DP with respect to the sensitive attributes. Then, the server aggregates these noisy sensitive gradients and the noiseless non-sensitive gradients $\nabla_\theta \ell(x, y, \theta)$ and updates the model parameters $\theta_{t+1}$ and $W_{t+1}$ by taking descent and ascent steps.

**Theorem 6.** *Let $\varepsilon \leqslant 2\ln(1/\delta)$, $\delta \in (0,1)$, m be the batch-size, and $T \geqslant \left(\tilde{n}\frac{\sqrt{\varepsilon}}{2m}\right)^2$. Assume $\mathcal{F}(x, \cdot)$ is $L_f$-Lipschitz for all x, and $|(W_t)_{r,q}| \leqslant R$ for all $t \in [T], r \in [k], q \in [l]$. Then, for $\sigma_w^2 \geqslant \frac{16T\ln(1/\delta)}{\varepsilon^2 \tilde{n}^2 \rho}$ and $\sigma_\theta^2 \geqslant \frac{16L_f^2 R^2 \ln(1/\delta)T}{\varepsilon^2 \tilde{n}^2 \rho}$, Algorithm 1 is $(\varepsilon, \delta)$-ISRL-DP with respect to the sensitive attributes for all data sets containing at least $\rho$-fraction of minority attributes.*

See Appendix C for the proof. Next, we provide a convergence guarantee for Algorithm 1.

NEW
[qZng]
FIX
[qZng]
FIX
[ZGVZ]

FIX
[qZng]

**Theorem 7.** *Assume $\ell(\cdot, x, y)$ and $\mathcal{F}(x, \cdot)$ are Lipschitz and have Lipschitz gradients. Then, there exist algorithmic parameters such that Algorithm 1 returns a $\hat{\theta}_T$ which is $(\epsilon, \delta)$-ISRL-DP with*

$$\mathbb{E}\|\nabla \mathit{FERMI}(\hat{\theta}_T)\|^2 = \mathcal{O}\left(\frac{\sqrt{\max(d_\theta, kl)\ln(1/\delta)}}{\varepsilon\tilde{n}\sqrt{N}}\right).$$

Note that we choose $T$ to achieve the best accuracy that our algorithm can achieve. We will clarify the choice of $T$ that implies the accuracy result in the above result. Compared to the central DP stationarity gap bound obtained in Lowy et al. (2023) (with $n = \tilde{n}N$), the bound in 7 is larger by a factor of $\sqrt{N}$. This is because ISRL-DP is a stronger privacy notion than central DP (Lowy & Razaviyayn, 2023) and our analysis accounts for *data heterogeneity* across silos.

Theorem 7 establishes that the proposed ISRL-DP algorithm converges to an approximate stationary point of the fairness-regularized objective, demonstrating that introducing the $\chi^2$-based fairness regularizer preserves provable convergence under differential privacy. It is important to note that Theorem 7 is not a direct guarantee on the demographic parity or equalized odds violation of the returned predictor, which would generally require additional assumptions relating stationarity to global optimality. The fairness improvements are evaluated empirically through the fairness–accuracy tradeoff curves.

NEW [qZng]

The proof of Theorem 14 follows from careful tracking of noise variance and sampling of data obtained from the different silos. A key observation is that even though the sampling is distributed across silos, the expected value of gradient after this modified form of sampling is an unbiased estimator of the global loss function due to linearity of expectation. Moreover, by averaging silos' noisy gradients, we reduce the total privacy noise variance.

While our approach leverages the DP min-max optimization techniques of Lowy et al. (2023), extending this framework to FL has its own challenges. Generally, each silo computes the sensitive-gradient terms locally, adds Gaussian noise before communication, and the server only aggregates the noisy sensitive gradients together with the non-sensitive loss gradients. The analysis must therefore account for distributed mini-batch sampling across silos, independent noise added at each silo, and the stronger transcript-level ISRL-DP guarantee. In particular, the sampling is distributed across silos with different data, which introduces additional challenges in analysis of the central updates. However, the expected value of gradient after this modified form of sampling is an unbiased estimator of the global loss function due to linearity of expectation helping us to overcome this challenge and derive bounds on sampling as to that of Lowy et al. (2023). See Appendix D for the detailed proof. In fact, in Appendix D, we prove Theorem 14, which is a general result that applies to *all smooth non-convex strongly-concave min-max optimization problems*, being of independent interest to the private optimization and federated learning community. We also extend the analysis for Algorithm 1 assuming all features to be sensitive in Appendix C.1.

NEW [qZng, ZGVZ]

**Interpretation of the fairness guarantee.** It is important to note that Theorem 7 should be interpreted as a convergence guarantee for the fairness-regularized objective FERMI, not as a direct bound on the demographic parity or equalized odds violation of the returned predictor. The $\chi^2$-regularizer is a surrogate for these fairness violations: when the regularizer is small, the corresponding fairness violation is small. However, because our theorem establishes approximate stationarity in a nonconvex problem, it does not by itself imply a small value of the fairness violation.

In Algorithm 1, we implicitly assume that the *frequency* of each sensitive attribute is known in order to compute $\hat{P}_S$ and broadcast it to the silos. This assumption is not very restrictive: In practice, releasing the frequency of the sensitive attributes of data is very common. Moreover, it is straightforward to privately estimate $\hat{P}_S$ using DP histograms. Thus, for simplicity, we assumed $\hat{P}_S$ to be known.

In the next section, we show our framework extends to *hybrid* centralization settings.

# 4 Different Modes of Data Centralization

Recall that we have assumed each silo is divided into two distinct parts: one that holds the *sensitive* data and another that holds *non-sensitive* data. The two divisions within each silo can communicate with each other and with all the sensitive and non-sensitive divisions of other silos. Leveraging this subtlety, we show how to model a wide range of hybrid centralized/distributed learning tasks that involve privacy of sensitive attributes. We will illustrate how Algorithm 1 readily extends to these hybrid tasks.

**One Silo, Centralized Sensitive and Non-Sensitive Data in Separate Subdivisions.** An example of this can be seen in healthcare organizations where the sensitive part of data can only be accessed by authorized personnel. In this case, we have $N = 1$ silo in SteFFLe. The updates from the sensitive subdivision are private due to the ISRL-DP guarantee in Theorem 6. Theorem 7 recovers the stationarity bound in Lowy et al. (2023).

**Centralized Sensitive Data and Decentralized Non-Sensitive Data.** In this case, we have 1 silo containing sensitive features and $N$ silos containing non-sensitive features. Our algorithm can be used to train models in this setting: in round $t$, instead of querying silo $i$'s sensitive division, the central server queries the central sensitive silo and receives noisy ISRL-DP sensitive gradients. These noisy sensitive gradients are combined with the noiseless non-sensitive gradients from each of the non-sensitive silos, and then the model is updated. An example of a silo containing centralized sensitive data is the *United States Census Bureau*. It provides essential demographic, social, and economic data that various institutions utilize for a wide range of purposes. Some examples of these institutions include government agencies, academics, and non-profit organizations. The data with these institutions correspond to silos with public data and any machine learning model they train for decision making would require a combination of their own data and the centralized sensitive data provided by the Census.

**General Case: Arbitrary Numbers of Sensitive and Non-Sensitive Silos.** Recall that every data-point we have is represented by a tuple $\{((x_u, y_u), s_u)\}_{u=1}^n$. We refer to $(x_u, y_u)$ as the non-sensitive part of the datapoint and $s_u$ to be the sensitive part. We assume that every silo indexes the data by universal index assigned to each data-point (say indexed by $1, 2, .., n$) instead of their local index. The data is distributed between the silos as follows:

- Let there be $p$ silos (represented by $1, ..., p$) with non-sensitive parts of the data (non-sensitive silos) and $s$ silos (represented by $1, ..., s$) with the sensitive parts of data (sensitive silos).

- Let $i \in [p]$ be the non-sensitive silo containing the non-sensitive attributes of datapoints which have indices $P_i \subset [n]$ such that $P_i \cap P_j = \phi$ for all $i, j \in [p]$ for $i \neq j$ and $\bigcup_{i=1}^p P_i = [n]$.

- Similarly, let any sensitive silo $i \in [s]$, contain the non-sensitive attributes of datapoints which have indices $S_i \subset [n]$ such that $S_i \cap S_j = \phi$ for all $i, j \in [s]$ for $i \neq j$ and $\bigcup_{i=1}^s S_i = [n]$.

For the training to happen, the non-sensitive silos locally sample a batch of data $\mathcal{J}_j \subset P_j$ for all $j \in [p]$. They compute the gradients of the loss with using their part of data and broadcast it to the server. For the gradient of the regularizer, each non-sensitive silo $j$ broadcasts $\mathcal{J}_j$ along with their respective model outputs. Then, the sensitive silos $c$ (such that $S_c \bigcap \bigcup_{j=1}^p \mathcal{J}_j \neq \varnothing$) which have the data corresponding to the indices sampled by all the non-sensitive silos $(S_c \bigcap \bigcup_{j=1}^p \mathcal{J}_j)$ compute the gradient using the broadcasted outputs by the model and their local sensitive data. Then, these silos locally add noise according to batch size being $|S_c \bigcap \bigcup_{j=1}^p \mathcal{J}_j|$, and broadcast these noisy gradients to the server. The server aggregates both gradients from the non-sensitive and the sensitive silos and updates the model parameters.

It is important that for every sensitive silo $c$ scales its noise according to batch size being $|S_c \bigcap \bigcup_{j=1}^p \mathcal{J}_j|$ to preserve ISRL DP. However, since we have assumed that only the sensitive silos corresponding to the data will participate implying that $|S_c \bigcap \bigcup_{j=1}^p \mathcal{J}_j| \geq 1$. Hence, the upper bound on the stationarity gap would still exist with the value of batch size being one, thus still preserving a theoretical guarantee.

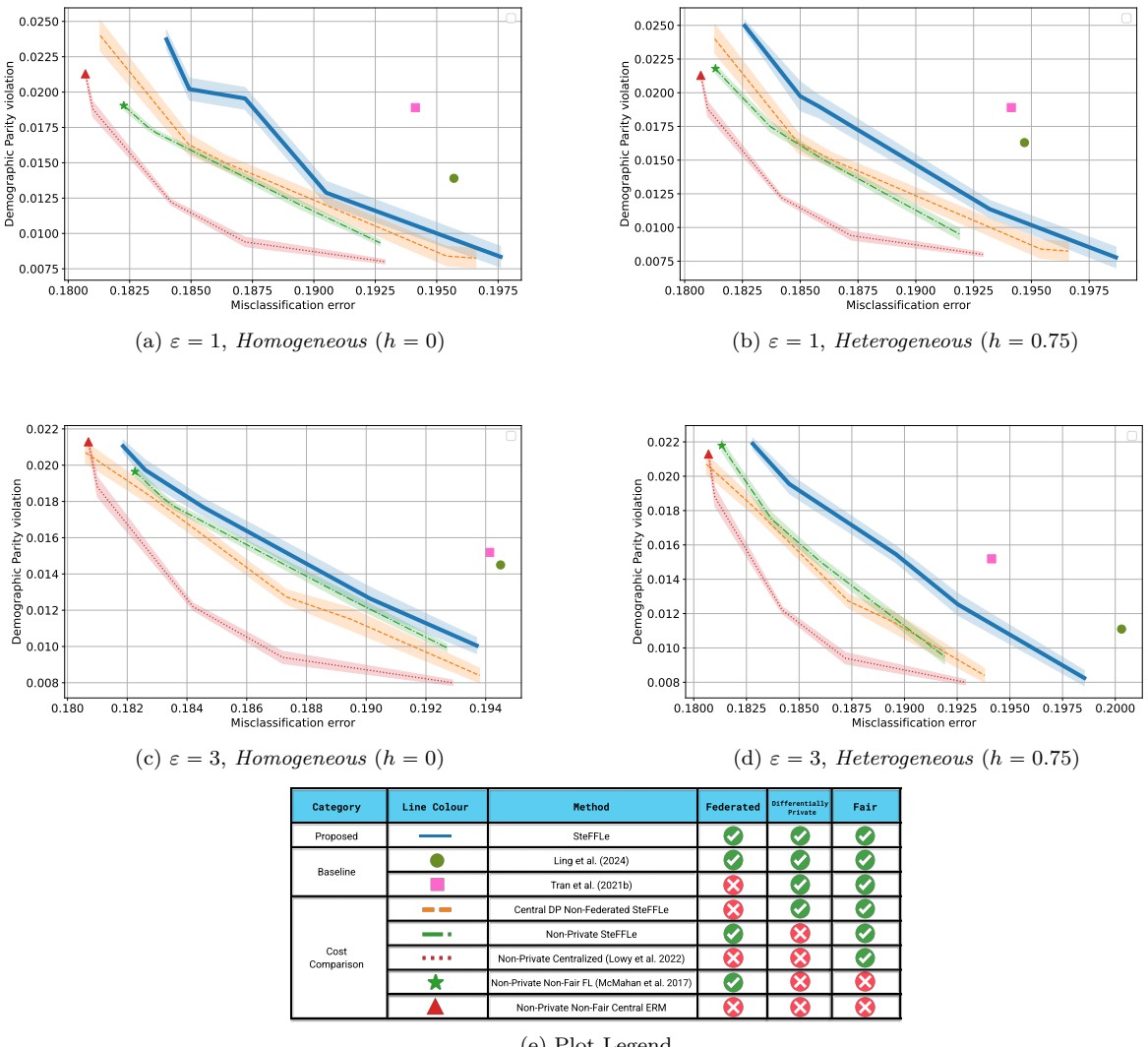

(a) $\varepsilon = 1$, *Homogeneous* ($h = 0$)

(b) $\varepsilon = 1$, *Heterogeneous* ($h = 0.75$)

(c) $\varepsilon = 3$, *Homogeneous* ($h = 0$)

(d) $\varepsilon = 3$, *Heterogeneous* ($h = 0.75$)

| Category | Line Colour | Method | Federated | Differentially Private | Fair |
|---|---|---|---|---|---|
| Proposed | —— | SteFFLe | ✓ | ✓ | ✓ |
| Baseline | ● | Ling et al. (2024) | ✓ | ✓ | ✓ |
| | ■ | Tran et al. (2021b) | ✗ | ✓ | ✓ |
| Cost Comparison | – – | Central DP Non-Federated SteFFLe | ✗ | ✓ | ✓ |
| | – · – | Non-Private SteFFLe | ✓ | ✗ | ✓ |
| | ···· | Non-Private Centralized (Lowy et al. 2022) | ✗ | ✗ | ✓ |
| | ★ | Non-Private Non-Fair FL (McMahan et al. 2017) | ✓ | ✗ | ✗ |
| | ▲ | Non-Private Non-Fair Central ERM | ✗ | ✗ | ✗ |

(e) Plot Legend

Figure 1: Demographic parity vs Misclassification error on *Credit Card* dataset (*Number of Silos* $= 3$)

**Unequal silo sizes.** For simplicity, we stated the main theorem for the case where each silo has $\tilde{n} = n/N$ samples. The result extends directly to unequal silo sizes. Let $n_j = |Z_j|$, $n = \sum_{j=1}^{N} n_j$, and $\alpha_j = n_j/n$. In this case, we optimize the sample-weighted objective

$$F(\theta, W) = \sum_{j=1}^{N} \alpha_j F_j(\theta, W), \qquad F_j(\theta, W) = \frac{1}{n_j} \sum_{i=1}^{n_j} f(\theta, W; z_{j,i}),$$

and the server aggregates silo gradients using weights $\alpha_j$ rather than $1/N$. The resulting stochastic gradient remains unbiased for the global empirical objective. The privacy proof is unchanged except that the Gaussian noise is calibrated separately for each silo using its local sample size $n_j$. Specifically, the noise scales as

$$\sigma_{j,W}^2 \geq \frac{T \log(1/\delta)}{\varepsilon^2 n_j^2 \rho}, \qquad \sigma_{j,\theta}^2 \geq \frac{L_\theta^2 D^2 T \log(1/\delta)}{\varepsilon^2 n_j^2 \rho}.$$

The convergence proof then follows by replacing the equal-size variance terms by the corresponding weighted sums. In particular, when mini-batch sizes are chosen proportionally to $n_j$, the stationarity bound has the

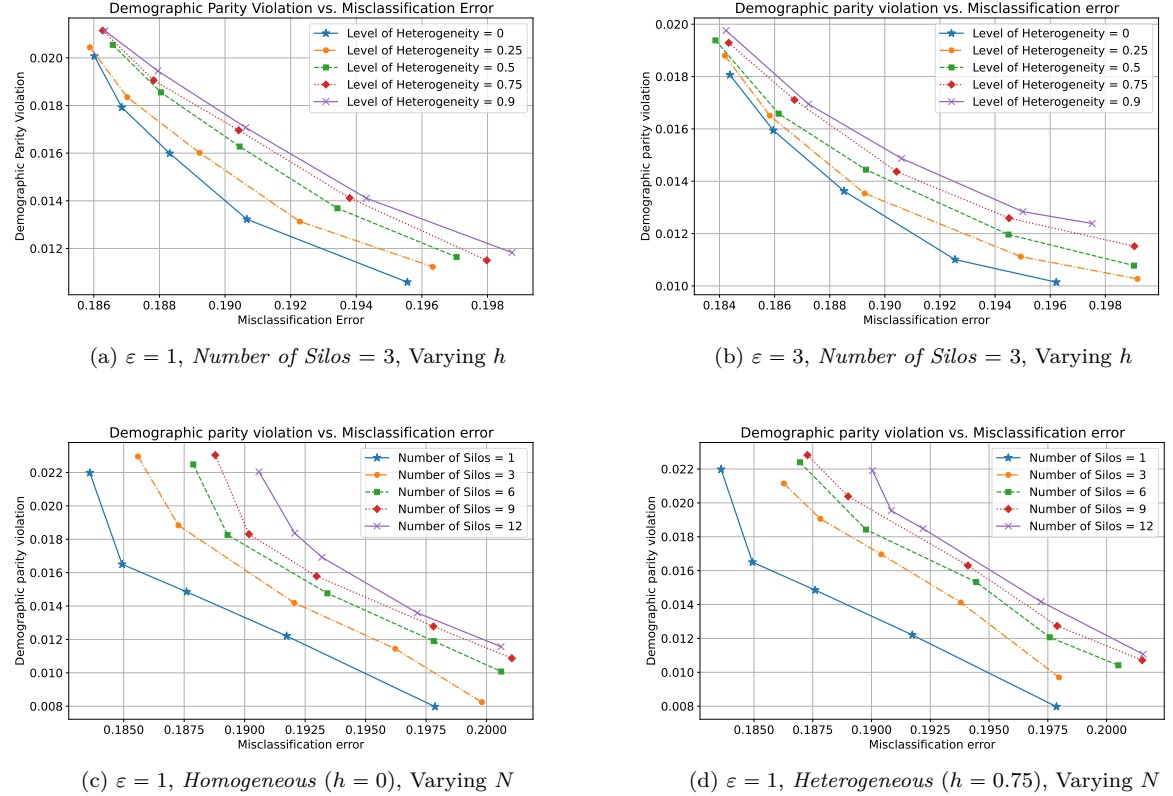

(a) $\varepsilon = 1$, *Number of Silos* $= 3$, Varying $h$   (b) $\varepsilon = 3$, *Number of Silos* $= 3$, Varying $h$

(c) $\varepsilon = 1$, *Homogeneous* $(h = 0)$, Varying $N$   (d) $\varepsilon = 1$, *Heterogeneous* $(h = 0.75)$, Varying $N$

Figure 2: Varying Levels of Heterogeneity ($h$) and Number of Silos ($N$) on the *Credit Card* dataset

same form as Theorem 7 with $\tilde{n}\sqrt{N}$ replaced by $n/\sqrt{N}$. Moreover, the condition for satisfying moments accountant then becomes $T \geqslant \max_{i \in [N]} \left( n_i \frac{\sqrt{\varepsilon}}{2m_i} \right)^2$.

NEW
[ZGVZ]

## 5 Numerical Experiments

In this section, we evaluate the performance of our algorithm in terms of fairness violation vs. test error for different levels of privacy, levels of silo heterogeneity, and numbers of silos. We present our results in two parts: In Section 5.1, we assess the performance of our method in training logistic regression models on several benchmark tabular datasets. In Section 5.2, we discuss how the fairness-accuracy tradeoffs are affected by silo heterogeneity and by the number of silos for a fixed privacy level.

**Average results.** To evaluate the overall performance of our algorithm and the existing baselines, we calculated the performance gain with respect to fairness violation (for fixed accuracy level) that our model yields *over all the datasets*. We obtained reductions in demographic parity violations of around 75.47% and 52.93% compared with Tran et al. (2021) and Ling et al. (2024). Note that the algorithm of Tran et al. (2021) is *not ISRL-DP*, instead satisfying only the weaker notion of central DP. We also obtained an average reduction in equalized odds violation of 95.42% compared to Ling et al. (2024). We specify our experimental setup, datasets, methods and additional results that we compare against in Appendix E.

### 5.1 Federated, Private, and Fair Logistic Regression

In the first set of experiments we train a logistic regression model using SteFFLe (Algorithm 1) to promote demographic parity. We compare SteFFLe against all applicable publicly available baselines in each exper-

iment. We carefully tuned the hyperparameters of all baselines for fair comparison. We find that *SteFFLe consistently outperforms all state-of-the-art baselines across all data sets in all privacy and heterogeneity levels.*

**Baselines.** The baselines include: (1) the approach by Tran et al. (2021), which is *central* differentially private and fair but *not federated and not ISRL-DP*; (2) the method of Ling et al. (2024), which incorporates federated learning, ISRL-DP, and fairness. These were the only DP fair baselines with code made publicly available for each experiment.

Additionally, we examine the *cost of incorporating federated learning and ISRL-DP* by measuring the fairness-accuracy trade-offs for different *variations of SteFFLE*. These variations include: *Central DP SteFFLe* Lowy et al. (2023), which is not ISRL-DP or federated, but still satisfies the weaker central DP notion and still promotes fairness; *Non-Private SteFFLe*, which is fair and federated, but not private; *Non-Private Centralized* Lowy et al. (2022a), which is fair, but not private or federated; *Non-Private Non-Fair FL* McMahan et al. (2018), which uses federated averaging; and *Non-Private Non-Fair Central ERM*, which simply uses SGD. See Figure 1 and the legend therein for our results on *Credit Card* dataset.

**Datasets.** We use three benchmark tabular datasets: Credit-Card, Adult Income, and Retired Adult dataset from the UCI machine learning repository (Dua & Graff (2017)). The predicted variables and sensitive attributes are both binary in these datasets. We analyze fairness-accuracy trade-offs with three different privacy budgets $\varepsilon \in \{1, 3, 9\}$ and two different values of heterogeneity levels $h = 0$ (homogeneous setting) and $h = 0.75$ (heterogeneous setting), keeping the number of silos $N = 3$ for each dataset. Note that these values of $\varepsilon$ and heterogeneity levels are standard in the literature for empirically comparing different algorithms in private and federated learning methods Lowy et al. (2023); Ghoukasian & Asoodeh (2024). We compare against state-of-the-art algorithms proposed in Ling et al. (2024) and (the demographic parity objective of) Tran et al. (2021). The results displayed are averages over 15 trials (random seeds) for each value of $\varepsilon, h$ and $N$.

**Results for different datasets.** Selected results for private and fair federated logistic regression on the Credit Card dataset are shown in Fig. 1. The remaining results of the Credit Card dataset and experiments of Adult and Retired Adult dataset are shown in Appendix E.5.1 and Appendix E.5.2. For logistic regression with equalized odds as the fairness violation, we provide further results (for a modified version of SteFFLe) on the Credit Card dataset in Appendix E.1. Compared to the baselines Tran et al. (2021) and Ling et al. (2024), SteFFLe offers superior fairness-accuracy tradeoffs at all privacy ($\varepsilon$) and heterogeneity levels ($h$) across all three datasets. Moreover, the method of Tran et al. (2021) is not ISRL-DP.

**Training on Large scale Datasets** In our second set of experiments, we train a large classifier ($d \approx 11.68$ million) on the UTKFace dataset consisting of 20,000 images. The classifier categorizes facial images into nine age groups, following a setup similar to Tran et al. (2022), while treating race (with five classes) as the sensitive attribute. Our results clearly demonstrate that *Algorithm 1 converges in a non-binary classification setting with small batch sizes and non-binary sensitive attributes.* Further details about the experiments, numerical results and observations can be found in Appendix E.3.

## 5.2 Impact of Silo Heterogeneity and the Number of Silos

In this section, we analyze the impact on SteFFLe's performance due to *varying heterogeneity levels* and *the number of silos* on the fairness-error trade-off, with a fixed privacy budget of $\varepsilon = 1$. We analyze how these factors affect demographic parity violation and misclassification error on the Credit Card dataset, as depicted in Fig. 2.

**Heterogeneous silo data is challenging in private fair FL.** We conducted experiments with silo heterogeneity levels ranging from 0 to 0.9, with 0 being homogeneous and 1 being heterogeneous. In Fig.2(a) and 2(b), the results demonstrate a *clear increase in both misclassification error and demographic parity violation as heterogeneity increases*, for a fixed number of $N = 3$ silos. This indicates that higher silo heterogeneity exacerbates the model's difficulty in achieving an optimal balance between fairness and accuracy.

Fig. 2(c) and 2(d) illustrates the effect of the number of silos on performance in both the homogeneous and heterogeneous settings. We vary the number of silos between $N \in [1, 12]$. In the homogeneous settings, as the number of silos increases from 1 to 12, both demographic parity violation rise and misclassification error grows. A similar trend is apparent in the heterogeneous setting, where an increase in the number of silos results in a proportional rise in both demographic parity violation and misclassification error in Fig. 2 (d). These findings suggest that *increasing the number of silos amplifies the challenges of maintaining fairness and accuracy, particularly under federated learning frameworks which incorporate privacy constraints.*

## 6 Conclusion and Discussion

Motivated by pressing ethical and legal concerns, we considered the problem of training fair and private ML models with decentralized data. We developed an algorithm that satisfies the strong ISRL-DP guarantee. We proved that our ISRL-DP algorithm converges for any minibatch size, without requiring (strong) convexity of the loss function. Finally, numerical experiments on several benchmark fairness data sets demonstrated that our method offers substantial fairness-accuracy benefits over the prior art, across different levels of privacy and silo heterogeneity. Our experiments also highlighted the challenges of silo heterogeneity for fair and accurate ISRL-DP FL. We hope this work inspires further research in private and fair federated learning. Future directions include exploring fundamental trade-offs between ISRL-DP, fairness, and accuracy, as well as improving performance in heterogeneous settings.

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

## A    Demographic Parity and Equalized Odds Version of ERMI

If demographic parity (Dwork et al., 2012) is the desired fairness notion, then one should use the following definition of Chi-Squared divergence as a regularizer Lowy et al. (2022a):

$$\widehat{D}_R(\widehat{Y}, S) := \sum_{j \in [l]} \sum_{r \in [k]} \frac{\hat{p}_{\widehat{Y},S}(j,r)^2}{\hat{p}_{\widehat{Y}}(j)\hat{p}_S(r)} - 1 \tag{5}$$

For equalized odds (Hardt et al., 2016b), one should use the following expression as a regularizer Lowy et al. (2022a):

$$\widehat{D}_R(\widehat{Y}; S|Y) := \mathbb{E}\left\{ \frac{\hat{p}_{\widehat{Y},S|Y}(\widehat{Y}, S|Y)}{\hat{p}_{\widehat{Y}|Y}(\widehat{Y}|Y)\hat{p}_{S|Y}(S|Y)} \right\} - 1$$

$$= \sum_{y=1}^{l} \sum_{j=1}^{l} \sum_{r=1}^{k} \frac{\hat{p}_{\widehat{Y},S|Y}(j,r|y)^2}{\hat{p}_{\widehat{Y}|Y}(j|y)\hat{p}_{S|Y}(r|y)} \hat{p}_Y(y) - 1. \tag{6}$$

In particular, if $D_R(\widehat{Y}; S|Y) = 0$, then $\widehat{Y}$ and $S$ are conditionally independent given $Y$ (i.e. equalized odds is satisfied).

## B    Complete Version of Theorem 5

Let $\widehat{\mathbf{y}}(x_{ji};\theta) \in \{0,1\}^l$ and $\mathbf{s}_{ji} \in \{0,1\}^k$ be the one-hot encodings of $\widehat{y}(x_{ji},\theta)$ and $s_{ji}$, respectively: i.e., $\widehat{\mathbf{y}}_h(x_i;\theta) = \mathbb{1}_{\{\widehat{y}(x_{ji},\theta)=h\}}$ and $\mathbf{s}_{ji,r} = \mathbb{1}_{\{s_{ji}=r\}}$ for $h \in [l], r \in [k]$. Also, denote $\widehat{P}_s = \text{diag}(\widehat{p}_S(1),\ldots,\widehat{p}_S(k))$, where $\widehat{p}_S(r) := \frac{1}{N\tilde{n}} \sum_{j=1}^{N} \sum_{i=1}^{\tilde{n}} \mathbb{1}_{\{s_{ji}=r\}} \geq \rho > 0$ is the empirical probability of attribute $r$ ($r \in [k]$). Then we have the following re-formulation of (FERMI obj.) as a min-max problem:

**Theorem 8** (Lowy et al. (2022a)). (FERMI obj.) *is equivalent to*

$$\min_{\theta} \max_{W \in \mathbb{R}^{k \times l}} \left\{ \widehat{F}(\theta, W) := \widehat{\mathcal{L}}(\theta) + \lambda \frac{1}{N\tilde{n}} \sum_{j=1}^{N} \sum_{i=1}^{\tilde{n}} \widehat{\psi}_{ji}(\theta, W) \right\}. \tag{7}$$

*where*

$$\widehat{\psi}_{ji}(\theta, W) := -\text{Tr}(W\mathbb{E}[\widehat{\mathbf{y}}(x_{ji},\theta)\widehat{\mathbf{y}}(x_{ji},\theta)^T|x_i]W^T)$$
$$+ 2\text{Tr}(W\mathbb{E}[\widehat{\mathbf{y}}(x_{ji};\theta)\mathbf{s}_{ji}^T|x_{ji},\mathbf{s}_{ji}]\widehat{P}_s^{-1/2}) - 1,$$

$\mathbb{E}[\widehat{\mathbf{y}}(x_{ji};\theta)\widehat{\mathbf{y}}(x_{ji};\theta)^T|x_{ji}] = \text{diag}(\mathcal{F}_1(x_{ji},\theta),\ldots,\mathcal{F}_l(x_{ji},\theta))$, *and* $\mathbb{E}[\widehat{\mathbf{y}}(x_{ji};\theta)\mathbf{s}_{ji}^T|x_{ji},\mathbf{s}_{ji}]$ *is a* $k \times l$ *matrix with* $\mathbb{E}[\widehat{\mathbf{y}}(x_{ji};\theta)\mathbf{s}_{ji}^T|x_{ji},\mathbf{s}_{ji}]_{r,u} = \mathbf{s}_{ji,r}\mathcal{F}_u(x_{ji},\theta)$.

Strong concavity of $\widehat{\psi}_{ji}$ is shown in Lowy et al. (2022a).

We also give the gradients of the estimated regularizer with respect to $\theta$ and $W$.

**Lemma 9.** *Given the above expression for $\psi_{ji}$,*

$$\nabla_\theta \widehat{\psi}_{ji}(\theta, W) = -\nabla_\theta \text{vec}\big(\mathbb{E}[\widehat{y}(x_{ji},\theta)\widehat{y}(x_{ji},\theta)^T \mid x_{ji}]\big)^T \text{vec}(W^T W) + 2\nabla_\theta \text{vec}\big(\mathbb{E}[s_{ji}\widehat{y}(x_{ji},\theta)^T \mid x_{ji}, s_{ji}]\big) \text{vec}\Big(W^T \widehat{P}_S^{-1/2}\Big),$$

*and*

$$\nabla_W \widehat{\psi}_{ji}(\theta, W) = -2W\mathbb{E}\big[\widehat{y}(x_{ji},\theta)\widehat{y}(x_{ji},\theta)^T \mid x_{ji}\big] + 2\widehat{P}_S^{-1/2}\mathbb{E}\big[s_{ji}\widehat{y}(x_{ji},\theta)^T \mid x_{ji}, s_{ji}\big].$$

NEW
[qZng]

# C  SteFFLe Algorithm: Privacy

To the privacy part of prove Theorem 6, we first consider the following definitions. For a particular silo $j$ at a particular iteration t, let the batch be denoted by $|B_t|$ and $\Delta_w^j$ denote the L2 sensitivity of gradient updates in silo $j$ with respect to $\theta$ and $w$ respectively.

$$\Delta_\theta^j = \sup_{Z_j \sim Z_j', \theta, W} \left\| \frac{1}{m} \sum_{i \in B_t} \left[ \nabla_\theta \widehat{\psi}(\theta, W; z_{ji}) - \nabla_\theta \widehat{\psi}(\theta, W; z_{ji}') \right] \right\|,$$

where we write $Z_j \sim Z_j'$ to mean that $Z_j$ and $Z_j'$ are two data sets (both with $\rho$-fraction of minority attributes) that differ in exactly one person's sensitive data. Likewise,

$$\Delta_W^j = \sup_{Z_j \sim Z_j', \theta, W} \left\| \frac{1}{m} \sum_{i \in B_t} \left[ \nabla_W \widehat{\psi}(\theta, W; z_{ji}) - \nabla_W \widehat{\psi}(\theta, W; z_{ji}') \right] \right\|.$$

Note that the per-silo sensitivity exactly reduces to the sensitivity in the centralized setting of Lowy et al. (2023) since the regularizer admits the same expression. Hence, we can use the sensitivity bounds in Lowy et al. (2023) to upper bound the sensitivity of the above quantities.

FIX [qZng]

**Lemma 10** (Lowy et al. (2023))**.** *With the above definition of neighbouring databases $Z$ and $Z'$ and assuming that $|W_{i,j}| \leqslant R$ for all $i \in [k]$ and $j \in [l]$ and $\mathcal{F}(x, \cdot)$ is $L_f$-Lipschitz for all $x$,*

$$\sup_{Z \sim Z', \theta, W} \left\| \frac{1}{|B|} \sum_{i \in B} \left[ \nabla_\theta \widehat{\psi}(\theta, W; z_i) - \nabla_\theta \widehat{\psi}(\theta, W; z_i') \right] \right\|^2 \leqslant \frac{8R^2 L_f^2}{|B|^2 \rho},$$

*and*

$$\sup_{Z \sim Z', \theta, W} \left\| \frac{1}{|B|} \sum_{i \in B} \left[ \nabla_W \widehat{\psi}(\theta, W; z_i) - \nabla_W \widehat{\psi}(\theta, W; z_i') \right] \right\|^2 \leqslant \frac{8}{|B|^2 \rho},$$

*Proof of Theorem 6.* Since, we have to guarantee ISRL-DP, we require that the gradients broadcasted from each *silo* is private. Using the fact that Lemma 10 holds for any silo, we get that $(\Delta_\theta^j)^2 \leqslant \frac{8R^2 L_\theta^2}{|B_t|^2 \rho}$ and $(\Delta_W^j)^2 \leqslant \frac{8}{|B_t|^2 \rho}$. By the Moments Accountant Theorem 1 of Abadi et al. (2016) and the fact that each silo has a total of $\tilde{n}$ datapoints, the claim follows. $\qquad\square$

## C.1  Algorithm for All Features Sensitive

In this section, we discuss the algorithm when all features are assumed to be sensitive. We borrow the sensitivity analysis from Lowy et al. (2023) for the regularizer gradients when all features are sensitive.

**Lemma 11** (Lowy et al. (2023))**.** *With the definition of neighbouring databases $Z$ and $Z'$ when all features are assumed to be sensitive and assuming that $|W_{i,j}| \leqslant R$ for all $i \in [k]$ and $j \in [l]$ and $\mathcal{F}(x, \cdot)$ is $L_f$-Lipschitz for all $x$,*

$$\sup_{Z \sim Z', \theta, W} \left\| \frac{1}{|B|} \sum_{i \in B} \left[ \nabla_\theta \widehat{\psi}(\theta, W; z_i) - \nabla_\theta \widehat{\psi}(\theta, W; z_i') \right] \right\|^2 \leqslant \frac{4R^2 L_f^2}{|B|^2} \left( \frac{4}{\rho} + l^2 \right),$$

*and*

$$\sup_{Z \sim Z', \theta, W} \left\| \frac{1}{|B|} \sum_{i \in B} \left[ \nabla_W \widehat{\psi}(\theta, W; z_i) - \nabla_W \widehat{\psi}(\theta, W; z_i') \right] \right\|^2 \leqslant \frac{8}{|B|^2} \left( \frac{1}{\rho} + R^2 \right),$$

**Theorem 12.** *Algorithm 2 is $(\varepsilon, \delta)$-DP with respect to all features when ${\sigma_\theta^{(1)}}^2 \geqslant \frac{36R^2 C_\theta^2 \ln(1/\delta)}{\varepsilon^2 n^2}$, ${\sigma_\theta^{(2)}}^2 \geqslant \frac{144R^2 L_f^2 \ln(1/\delta)}{\varepsilon^2 n^2} \left( \frac{4}{\rho} + l^2 \right)$ and $\sigma_w^2 \geqslant \frac{288T \ln(1/\delta)}{\varepsilon^2 n^2} \left( \frac{1}{\rho} + R^2 \right)$.*

---

**Algorithm 2** SteFFLe: Stochastic Private Fair Federated Learning for all Private Features

---

1: **Input:** $\{Z_j = \{x_{j,i}, y_{j,i}\}_{i=1}^{\tilde{n}}, \{s_{j,i}\}_{i=1}^{\tilde{n}}\}_{j=1}^N$, $\theta_0 \in \mathbb{R}^{d_\theta}$, $W_0 = 0 \in \mathbb{R}^{k \times l}$, step-sizes $(\eta_\theta, \eta_w)$, fairness parameter $\lambda \geqslant 0$, iteration number $T$, minibatch size $|B_t| = m \in [\tilde{n}]$, set $\mathcal{W} \subset \mathbb{R}^{k \times l}$, noise parameters $\{\sigma_{j,w}^2, \sigma_{j,\theta}^2\}_{j=1}^N$.

2: Compute $\widehat{P}_S^{-1/2} = \mathrm{diag}(\widehat{p}_S(1)^{-1/2}, \ldots, \widehat{p}_S(k)^{-1/2})$, where $\widehat{p}_S(r) := \frac{1}{N\tilde{n}} \sum_{j=1}^N \sum_{i=1}^{\tilde{n}} \mathbb{1}_{\{s_{ji}=r\}} \geqslant \rho > 0$.

3: **for** $t = 0, 1, \ldots, T-1$ **do**

4:     Central server sends $\theta_t, W_t$ to all silos.

5:     **for** $j \in [N]$ **in parallel do**

6:         Silo $j$ draws a mini-batch $B_t$ of data points $\{(x_{j,i}, y_{j,i}), s_{j,i}\}_{i \in B_t}$.

7:         Silo $j$ computes stochastic gradient $g_{t,j} := \frac{1}{|B_t|} \sum_{i \in B_t} \mathrm{Clip}_{C_\theta}\left(\nabla_\theta \ell(x_{j,i}, y_{j,i}, \theta_t)\right) + s_{t,j}$ with $s_{t,j} \sim \mathcal{N}\left(0, \sigma_\theta^{(1)^2} \mathbb{I}_{d_\theta}\right)$ where $\ell(x_{j,i}, y_{j,i}, \theta_t) = \mathrm{loss}\left(\mathcal{F}(x_{j,i}, \theta_t), y_{j,i}\right)$ and $\mathcal{F}$ is the Neural Network used to compute the probabiliy distribution of the predicted label.

8:         Silo $j$ computes $\{\mathcal{F}(x_{j,i}, \theta_t), \mathrm{Clip}_{L_f}\left(\nabla \mathcal{F}(x_{j,i}, \theta_t)\right), j, i\}_{i \in B_t}$.

9:         Silo $j$ computes noisy sensitive stochastic gradients $h_{t,j,\theta} := \frac{1}{|B_t|} \sum_{i \in B_t} \nabla_\theta \widehat{\psi}_{j,i}(\theta_t, W_t) + u_{t,j}$ and $h_{t,j,w} := \frac{1}{|B_t|} \sum_{i \in B_t} \nabla_w \widehat{\psi}_{j,i}(\theta_t, W_t) + V_{t,j}$, where $u_{t,j} \sim \mathcal{N}(0, \sigma_\theta^{(2)^2} \mathbf{I}_{d_\theta})$ and $V_{t,j}$ is a $k \times l$ matrix with independent random Gaussian entries $(V_t)_{q,r} \sim \mathcal{N}(0, \sigma_w^2)$.

10:         Silo $j$ broadcasts $g_{t,j}$, $h_{t,j,\theta}$, and $h_{t,j,w}$ to the central server.

11:     **end for**

12:     Central server updates $\theta_{t+1} \leftarrow \theta_t - \frac{\eta_\theta}{N} \sum_{j=1}^N [g_{t,j} + \lambda h_{t,j,\theta}]$ and $W_{t+1} \leftarrow \Pi_{\mathcal{W}}\left(W_t + \frac{\lambda \eta_w}{N} \sum_{j=1}^N h_{t,j,w}\right)$ where $\Pi_{\mathcal{W}}(x) = \mathrm{argmin}_{y \in \mathcal{W}} ||x - y||_2$.

13: **end for**

14: Pick $\hat{t}$ uniformly at random from $\{1, \ldots, T\}$.

15: **Return:** $\hat{\theta}_T := \theta_{\hat{t}}$.

---

*Proof.* The proof follows exactly in the same way as that of Theorem 6 with the sensitivity calculations from Lemma 11. Moreover, the updates of the loss with respect to $\theta$ have to be updated as well and hence, using the fact that we are clipping using $C_\theta$ for $\nabla \ell_\theta$, we get that the squared sensitivity for the update on Line 7 of the algorithm is upper bounded by $4C_\theta^2$ and thus by Abadi et al. (2016), we get privacy with respect to the the collection across $T$ time-steps with the claimed noise. □

**Note:** It is important to note that we can also get DP with respect to all the features simply by instantiating Algorithm 3 and doing per-sample clipping for gradients of the complete loss function with respect to $\theta$ and $W$ respectively and get privacy guarantees. In the above section, we simply show that our framework can also be extended to the setting when all features are private.

The sensitivity computations only change the constants but the dependence on $d_\theta, k, l, \varepsilon, \delta$ and $n$ follow exactly from Theorem 14 and hence the utility guarantee for Algorithm 2 follows from Theorem 7 as well.

# D    SteFFLe Algorithm: Utility

To prove Theorem 7, we will first derive a more general result. Namely, in Appendix D.1, we will provide a precise upper bound on the stationarity gap of noisy ISRL-DP federated stochastic gradient descent ascent (ISRL-DP-Fed-SGDA). We majorly follow the approach used by Lowy et al. (2023) to derive the stationarity gap.

## D.1    Noisy ISRL-DP Fed-SGDA for Nonconvex-Strongly Concave Min-Max FL Problems

In this subsection, we will prove Theorem 14, which implies the utility guarantee claimed in Theorem 7 (which we re-state as a standalone theorem in Theorem 21).

Let $Z = (Z_1, \ldots, Z_N)$ be a distributed dataset with $Z_j = \{z_{j,i}\}_{i=1}^{\tilde{n}}$ for $j \in [N]$. Consider a generic (smooth) nonconvex-strongly concave min-max federated ERM problem:

$$\min_{\theta \in \mathbb{R}^{d_\theta}} \max_{w \in \mathcal{W}} \left\{ F(\theta, w) := \frac{1}{N\tilde{n}} \sum_{j=1}^{N} \sum_{i=1}^{\tilde{n}} f(\theta, w; z_{j,i}) \right\}, \tag{8}$$

where $f(\theta, \cdot; z)$ is $\mu$-strongly concave for all $\theta, z$ but $f(\cdot, w; z)$ is potentially non-convex. Grant Assumption 13.

---

**Algorithm 3** Noisy ISRL-DP Federated Stochastic Gradient Descent-Ascent (ISRL-DP-Fed-SGDA)

---

1: **Input**: data $Z$, $\theta_0 \in \mathbb{R}^{d_\theta}$, $w_0 \in \mathcal{W}$, step-sizes $(\eta_\theta, \eta_w)$, privacy noise parameters $\tilde{\sigma}_\theta, \tilde{\sigma}_w$, batch size $m$, iteration number $T \geqslant 1$.
2: **for** $t = 0, 1, \ldots, T-1$ **do**
3:     Central server communicates $(\theta_t, w_t)$ to silos.
4:     **for** $j \in [N]$ **in parallel do**
5:         Silo $j$ draws a fresh batch of data points $\{z_{j,i}\}_{i=1}^{m}$ uniformly at random from $Z_j$ with replacement.
6:         Silo $j$ draws fresh independent Gaussian noises $u_j \sim \mathcal{N}(0, \tilde{\sigma}_\theta^2 \mathbf{I}_{d_\theta})$ and $v_j \sim \mathcal{N}(0, \tilde{\sigma}_w^2 \mathbf{I}_{d_w})$.
7:         Silo $j$ communicates noisy stochastic gradients $H_{j,\theta} = \frac{1}{m} \sum_{i=1}^{m} \nabla_\theta f(\theta_t, w_t, z_{j,i}) + u_j$ and $H_{j,w} = \frac{1}{m} \sum_{i=1}^{m} \nabla_w f(\theta_t, w_t, z_{j,i}) + v_j$ to server.
8:     **end for**
9:     Central server aggregates noisy stochastic gradients and updates the global model: $\theta_{t+1} \leftarrow \theta_t - \eta_\theta \left( \frac{1}{N} \sum_{j=1}^{N} H_{j,\theta} \right)$ and $w_{t+1} \leftarrow \Pi_{\mathcal{W}} \left[ w_t + \eta_w \left( \frac{1}{N} \sum_{j=1}^{N} H_{j,\theta} \right) \right]$.
10: **end for**
11: Draw $\hat{\theta}_T$ uniformly at random from $\{\theta_t\}_{t=1}^{T}$.
12: **Return:** $\hat{\theta}_T$.

---

**Assumption 13.**     1. $f(\cdot, w; z)$ is $L_\theta$-Lipschitz and $\beta_\theta$-smooth for all $w \in \mathcal{W}, z \in \mathcal{Z}$.

2. $f(\theta, \cdot; z)$ is $L_w$-Lipschitz, $\beta_w$-smooth, and $\mu$-strongly concave on $\mathcal{W}$ for all $\theta \in \mathbb{R}^{d_\theta}$, $z \in \mathcal{Z}$.

3. $\|\nabla_w f(\theta, w; z) - \nabla_w f(\theta', w; z)\| \leqslant \beta_{\theta w} \|\theta - \theta'\|$ and $\|\nabla_\theta f(\theta, w; z) - \nabla_\theta f(\theta, w'; z)\| \leqslant \beta_{\theta w} \|w - w'\|$ for all $\theta, \theta', w, w', z$.

4. $\mathcal{W}$ has $\ell_2$ diameter bounded by $D \geqslant 0$.

5. $\nabla_w F(\theta, w^*(\theta)) = 0$ for all $\theta$, where $w^*(\theta) := \operatorname{argmax}_w F(\theta, w)$ is the unconstrained global maximizer.

We denote $\kappa_w := \frac{\beta_w}{\mu}$ and $\kappa_{\theta w} := \frac{\beta_{\theta w}}{\mu}$. Also, let

$$\Phi(\theta) := \max_{w \in \mathcal{W}} F(\theta, w).$$

We can now provide our general, precise privacy and utility guarantee for Algorithm 3:

**Theorem 14** (Privacy and Utility of Algorithm 3). *Let* $\varepsilon \leqslant 2\ln(1/\delta)$, $\delta \in (0,1)$. *Grant Assumption 13. Choose* $\sigma_w^2 = \frac{8TL_w^2 \ln(1/\delta)}{\varepsilon^2 \tilde{n}^2}$, $\sigma_\theta^2 = \frac{8TL_\theta^2 \ln(1/\delta)}{\varepsilon^2 \tilde{n}^2}$, *and* $T \geqslant \left( \tilde{n} \frac{\sqrt{\varepsilon}}{2m} \right)^2$. *Then Algorithm 3 is* $(\varepsilon, \delta)$-DP. *Further, if we choose* $\eta_\theta = \frac{1}{16\kappa_w (\beta_\theta + \beta_{\theta w}\kappa_{\theta w})}$, $\eta_w = \frac{1}{\beta_w}$, *and* $T \approx \sqrt{\kappa_w [\Delta_\Phi (\beta_\theta + \beta_{\theta w}\kappa_{\theta w}) + \beta_{\theta w}^2 D^2]} \varepsilon \tilde{n} \sqrt{N} \min \left( \frac{1}{L_\theta \sqrt{d_\theta}}, \frac{\beta_w}{\beta_{\theta w} L_w \sqrt{\kappa_w d_w}} \right)$, *then*

$$\mathbb{E}\|\nabla\Phi(\hat{\theta}_T)\|^2 \lesssim \sqrt{\Delta_\Phi (\beta_\theta + \beta_{\theta w}\kappa_{\theta w})\kappa_w + \kappa_w \beta_{\theta w}^2 D^2} \left[ \frac{L_\theta \sqrt{d_\theta} \ln(1/\delta)}{\varepsilon \tilde{n} \sqrt{N}} + \left( \frac{\beta_{\theta w} \sqrt{\kappa_w}}{\beta_w} \right) \frac{L_w \sqrt{d_w} \ln(1/\delta)}{\varepsilon \tilde{n} \sqrt{N}} \right]$$

$$+ \frac{\mathbb{1}_{\{m < \tilde{n}\}}}{mN} \left( L_\theta^2 + \frac{\kappa_w \beta_{\theta w}^2 L_w^2}{\beta_w^2} \right).$$

*In particular, if $m \geqslant \min\left(\frac{\varepsilon\tilde{n}L_\theta}{\sqrt{Nd_\theta\kappa_w[\Delta_\Phi(\beta_\theta+\beta_{\theta w}\kappa_{\theta w})+\beta_{\theta w}^2 D^2]}}, \frac{\varepsilon\tilde{n}L_w\sqrt{\kappa_w}}{\sqrt{N}\beta_{\theta w}\beta_w\sqrt{d_w\kappa_w[\Delta_\Phi(\beta_\theta+\beta_{\theta w}\kappa_{\theta w})+\beta_{\theta w}^2 D^2]}}\right)$, then*

$$\mathbb{E}\|\nabla\Phi(\hat{\theta}_T)\|^2 \lesssim \sqrt{\kappa_w[\Delta_\Phi(\beta_\theta + \beta_{\theta w}\kappa_{\theta w}) + \beta_{\theta w}^2 D^2]}\left(\frac{\sqrt{\ln(1/\delta)}}{\varepsilon\tilde{n}\sqrt{N}}\right)\left(L_\theta\sqrt{d_\theta} + \left(\frac{\beta_{\theta w}\sqrt{\kappa_w}}{\beta_w}\right)L_w\sqrt{d_w}\right).$$

The proof of Theorem 14 will require several technical lemmas. These technical lemmas, in turn, require some preliminary lemmas, which we present below.

**Lemma 15** (Lowy et al. (2023); Lin et al. (2020)). *Grant Assumption 13. Then $\Phi$ is $2(\beta_\theta + \beta_{\theta w}\kappa_{\theta w})$-smooth with $\nabla\Phi(\theta) = \nabla_\theta F(\theta, w^*(\theta))$, and $w^*(\cdot)$ is $\kappa_w$-Lipschitz.*

**Lemma 16** (Lei et al. (2017)). *Let $\{a_l\}_{l\in[n]}$ be an arbitrary collection of vectors such that $\sum_{l=1}^n a_l = 0$. Further, let $\mathcal{S}$ be a uniformly random subset of $[n]$ of size $m$. Then,*

$$\mathbb{E}\left\|\frac{1}{m}\sum_{l\in\mathcal{S}} a_l\right\|^2 = \frac{n-m}{(n-1)m}\frac{1}{n}\sum_{l=1}^n \|a_l\|^2 \leqslant \frac{\mathbb{1}_{\{m<n\}}}{m}\frac{1}{n}\sum_{l=1}^n \|a_l\|^2.$$

**Lemma 17** (Co-coercivity of the gradient). *For any $\beta$-smooth and convex function $g$, we have*

$$\|\nabla g(a) - \nabla g(b)\|^2 \leqslant 2\beta(g(a) - g(b) - \langle g(b), a - b\rangle),$$

*for all $a, b \in domain(g)$.*

**Lemma 18.** *For all $t \in [T]$, the iterates of Algorithm 3 satisfy*

$$\mathbb{E}\Phi(\theta_t) \leqslant \Phi(\theta_{t-1}) - \left(\frac{\eta_\theta}{2} - 2(\beta_\theta + \beta_{\theta w}\kappa_{\theta w})\eta_\theta^2\right)\|\nabla\Phi(\theta_{t-1})\|^2$$
$$+ \left(\frac{\eta_\theta}{2} + 2(\beta_\theta + \beta_{\theta w}\kappa_{\theta w})\eta_\theta^2\right)\|\nabla\Phi(\theta_{t-1}) - \nabla_\theta F(\theta_{t-1}, w_{t-1})\|^2 + (\beta_\theta + \beta_{\theta w}\kappa_{\theta w})\eta_\theta^2\left(d_\theta\frac{\sigma_\theta^2}{N} + \frac{4L_\theta^2}{m}\mathbb{1}_{\{m<\tilde{n}\}}\right),$$

*conditional on $\theta_{t-1}, w_{t-1}$.*

*Proof.* Let us denote $\tilde{g} := \frac{1}{N}\sum_{i=1}^N\left(\frac{1}{m}\sum_{i=1}^m \nabla_\theta f(\theta_{t-1}, w_{t-1}; z_{ji}) + u_{t-1,i}\right) := g + u_{t-1}$, so $\theta_t = \theta_{t-1} - \eta_\theta\tilde{g}$, where $u_{t-1} = \frac{1}{N}\sum_{i=1}^N u_{t-1,i}$. Since $u_{t-1,1}, u_{t-1,2}, ..., u_{t-1,N}$ are i.i.d. sampled from $\mathcal{N}(0, \sigma_\theta^2)$, we have that $u_{t-1} \sim \mathcal{N}(0, \frac{\sigma_\theta^2}{N})$. We proceed with the proof, conditioning on the randomness due to sampling and Gaussian noise addition. By smoothness of $\Phi$ (see Lemma 15), we have

$$\Phi(\theta_t) \leqslant \Phi(\theta_{t-1}) - \eta_\theta\langle\tilde{g}, \nabla\Phi(\theta_{t-1})\rangle + (\beta_\theta + \beta_{\theta w}\kappa_{\theta w})\eta_\theta^2\|\tilde{g}\|^2$$
$$= \Phi(\theta_{t-1}) - \eta_\theta\|\nabla\Phi(\theta_{t-1})\|^2 - \eta_\theta\langle\tilde{g} - \nabla\Phi(\theta_{t-1}), \nabla\Phi(\theta_{t-1})\rangle + (\beta_\theta + \beta_{\theta w}\kappa_{\theta w})\eta_\theta^2\|\tilde{g}\|^2.$$

Taking expectation (conditional on $\theta_{t-1}, w_{t-1}$) and using the fact that the Gaussian noise has mean zero and is independent of $(\theta_{t-1}, w_{t-1}, Z)$, plus the fact that $\mathbb{E}[g] = \nabla_\theta F(\theta_{t-1}, w_{t-1})$, we get

$$\mathbb{E}[\Phi(\theta_t)] \leqslant \Phi(\theta_{t-1}) - \eta_\theta\|\nabla\Phi(\theta_{t-1})\|^2 - \eta_\theta\langle\nabla_\theta F(\theta_{t-1}, w_{t-1}) - \nabla\Phi(\theta_{t-1}), \nabla\Phi(\theta_{t-1})\rangle$$
$$+ (\beta_\theta + \beta_{\theta w}\kappa_{\theta w})\eta_\theta^2\left[d_\theta\frac{\sigma_\theta^2}{N} + \underbrace{\mathbb{E}\|g - \nabla_\theta F(\theta_{t-1}, w_{t-1})\|^2}_{①} + \|\nabla_\theta F(\theta_{t-1}, w_{t-1})\|^2\right]$$

Expanding $\mathbb{E}\|g - \nabla_\theta F(\theta_{t-1}, w_{t-1})\|^2$ and using Young's inequality, we get that

$$
\begin{aligned}
① &= \mathbb{E}\left\|\frac{1}{N}\sum_{j=1}^{N}\left(\frac{1}{m}\sum_{i=1}^{m}\nabla_\theta f(\theta_{t-1}, w_{t-1}; z_{j,i})\right) - \frac{1}{N}\sum_{j=1}^{N}\left(\frac{1}{\tilde{n}}\sum_{k=1}^{\tilde{n}}\nabla_\theta f(\theta_{t-1}, w_{t-1}; z_{j,k})\right)\right\|^2 \\
&= \frac{1}{N^2}\mathbb{E}\left\|\sum_{j=1}^{N}\frac{1}{m}\sum_{i=1}^{m}\left(\nabla_\theta f(\theta_{t-1}, w_{t-1}; z_{j,i}) - \frac{1}{\tilde{n}}\sum_{k=1}^{\tilde{n}}\nabla_\theta f(\theta_{t-1}, w_{t-1}; z_{j,k})\right)\right\|^2 \\
&\leqslant \frac{1}{N}\sum_{j=1}^{N}\mathbb{E}\underbrace{\left\|\frac{1}{m}\sum_{i=1}^{m}\left(\nabla_\theta f(\theta_{t-1}, w_{t-1}; z_{j,i}) - \frac{1}{\tilde{n}}\sum_{k=1}^{\tilde{n}}\nabla_\theta f(\theta_{t-1}, w_{t-1}; z_{j,k})\right)\right\|^2}_{②}.
\end{aligned}
$$

For any silo $j$ , we get that

$$
\begin{aligned}
② &\leqslant \frac{\mathbb{1}_{\{m<\tilde{n}\}}}{m\,\tilde{n}}\sum_{l=1}^{\tilde{n}}\left\|\nabla_\theta f(\theta_{t-1}, w_{t-1}; z_{j,l}) - \frac{1}{\tilde{n}}\sum_{i=1}^{\tilde{n}}\nabla_\theta f(\theta_{t-1}, w_{t-1}; z_{j,i})\right\|^2 \\
&\leqslant \frac{\mathbb{1}_{\{m<\tilde{n}\}}}{m\,\tilde{n}}\sum_{l=1}^{\tilde{n}}\left\|\frac{1}{\tilde{n}}\sum_{i=1}^{\tilde{n}}\nabla_\theta f(\theta_{t-1}, w_{t-1}; z_{j,l}) - \frac{1}{\tilde{n}}\sum_{i=1}^{\tilde{n}}\nabla_\theta f(\theta_{t-1}, w_{t-1}; z_{j,i})\right\|^2 \\
&\leqslant \frac{\mathbb{1}_{\{m<\tilde{n}\}}}{m\,\tilde{n}^2}\sum_{l=1}^{\tilde{n}}\sum_{i=1}^{\tilde{n}}\|\nabla_\theta f(\theta_{t-1}, w_{t-1}; z_{j,l}) - \nabla_\theta f(\theta_{t-1}, w_{t-1}; z_{j,i})\|^2 \leqslant \frac{4\mathbb{1}_{\{m<\tilde{n}\}}L_\theta^2}{m}.
\end{aligned}
$$

We used Lemma 16 in the first inequality. In the second inequality, we used Young's inequality and Lipschitz continuity of $f$. The remainder of the proof follows from Lowy et al. (2023). We restate the further steps for ease. Thus, we get that

$$
\begin{aligned}
\mathbb{E}[\Phi(\theta_t)] &\leqslant \Phi(\theta_{t-1}) - \eta_\theta\|\nabla\Phi(\theta_{t-1})\|^2 - \eta_\theta\langle\nabla_\theta F(\theta_{t-1}, w_{t-1}) - \nabla\Phi(\theta_{t-1}), \nabla\Phi(\theta_{t-1})\rangle \\
&\quad + (\beta_\theta + \beta_{\theta w}\kappa_{\theta w})\eta_\theta^2\left[d_\theta\frac{\sigma_\theta^2}{N} + \frac{4L_\theta^2}{m}\mathbb{1}_{\{m<\tilde{n}\}} + \|\nabla_\theta F(\theta_{t-1}, w_{t-1})\|^2\right] \\
&\leqslant \Phi(\theta_{t-1}) - \eta_\theta\|\nabla\Phi(\theta_{t-1})\|^2 - \eta_\theta\langle\nabla_\theta F(\theta_{t-1}, w_{t-1}) - \nabla\Phi(\theta_{t-1}), \nabla\Phi(\theta_{t-1})\rangle \\
&\quad + (\beta_\theta + \beta_{\theta w}\kappa_{\theta w})\eta_\theta^2\left[d_\theta\frac{\sigma_\theta^2}{N} + \frac{4L_\theta^2}{m}\mathbb{1}_{\{m<\tilde{n}\}} + 2\|\nabla_\theta F(\theta_{t-1}, w_{t-1}) - \nabla\Phi(\theta_{t-1})\|^2 + 2\|\nabla\Phi(\theta_{t-1})\|^2\right] \\
&\leqslant \Phi(\theta_{t-1}) - \eta_\theta\|\nabla\Phi(\theta_{t-1})\|^2 + \frac{\eta_\theta}{2}\left[\|\nabla\Phi(\theta_{t-1}) - \nabla_\theta F(\theta_{t-1}, w_{t-1})\|^2 + \|\nabla\Phi(\theta_{t-1})\|^2\right] \\
&\quad + (\beta_\theta + \beta_{\theta w}\kappa_{\theta w})\eta_\theta^2\left[d_\theta\frac{\sigma_\theta^2}{N} + \frac{4L_\theta^2}{m}\mathbb{1}_{\{m<\tilde{n}\}} + 2\|\nabla_\theta F(\theta_{t-1}, w_{t-1}) - \nabla\Phi(\theta_{t-1})\|^2 + 2\|\nabla\Phi(\theta_{t-1})\|^2\right] \\
&\leqslant \Phi(\theta_{t-1}) - \left(\frac{\eta_\theta}{2} - 2(\beta_\theta + \beta_{\theta w}\kappa_{\theta w})\eta_\theta^2\right)\|\nabla\Phi(\theta_{t-1})\|^2 \\
&\quad + \left(\frac{\eta_\theta}{2} + 2(\beta_\theta + \beta_{\theta w}\kappa_{\theta w})\eta_\theta^2\right)\|\nabla\Phi(\theta_{t-1}) - \nabla_\theta F(\theta_{t-1}, w_{t-1})\|^2 \\
&\quad + (\beta_\theta + \beta_{\theta w}\kappa_{\theta w})\eta_\theta^2\left(d_\theta\frac{\sigma_\theta^2}{N} + \frac{4L_\theta^2}{m}\mathbb{1}_{\{m<\tilde{n}\}}\right).
\end{aligned}
$$

In the second and third inequalities, we used Young's inequality and Cauchy-Schwartz. $\qquad\square$

**Lemma 19.** *Grant Assumption 13. If $\eta_\theta = \frac{1}{16\kappa_w(\beta_\theta + \beta_{\theta w}\kappa_{\theta w})}$, then the iterates of Algorithm 3 satisfy ($\forall t \geqslant 0$)*

$$
\mathbb{E}\Phi(\theta_{t+1}) \leqslant \mathbb{E}\left[\Phi(\theta_t) - \frac{3}{8}\eta_\theta\|\Phi(\theta_t)\|^2 + \frac{5}{8}\eta_\theta\left(\beta_{\theta w}^2\|w^*(\theta_t) - w_t\|^2 + d_\theta\frac{\sigma_\theta^2}{N} + \frac{4L_\theta^2}{m}\mathbb{1}_{\{m<\tilde{n}\}}\right)\right].
$$

With some changes to the noise variance, the proof follows exactly from Lowy et al. (2023). We restate it here for ease.

*Proof.* By Lemma 18, we have

$$\mathbb{E}\Phi(\theta_{t+1}) \leqslant \mathbb{E}\Phi(\theta_t) - \left(\frac{\eta_\theta}{2} - 2(\beta_\theta + \beta_{\theta w}\kappa_{\theta w})\eta_\theta^2\right)\mathbb{E}\|\nabla\Phi(\theta_t)\|^2$$

$$+ \left(\frac{\eta_\theta}{2} + 2\eta_\theta^2(\beta_\theta + \beta_{\theta w}\kappa_{\theta w})\mathbb{E}\|\nabla\Phi(\theta_t) - \nabla_\theta F(\theta_t, w_t)\|^2\right) + (\beta_\theta + \beta_{\theta w}\kappa_{\theta w})\eta_\theta^2 \left(d_\theta \frac{\sigma_\theta^2}{N} + \frac{4L_\theta^2}{m}\mathbb{1}_{\{m<\tilde{n}\}}\right)$$

$$\leqslant \mathbb{E}\Phi(\theta_t) - \frac{3}{8}\eta_\theta \mathbb{E}\|\nabla\Phi(\theta_t)\|^2 + \frac{5}{8}\eta_\theta \left[\mathbb{E}\|\nabla\Phi(\theta_t) - \nabla_\theta F(\theta_t, w_t)\|^2 + d_\theta \frac{\sigma_\theta^2}{N} + \frac{4L_\theta^2}{m}\mathbb{1}_{\{m<\tilde{n}\}}\right]$$

$$\leqslant \mathbb{E}\Phi(\theta_t) - \frac{3}{8}\eta_\theta \mathbb{E}\|\nabla\Phi(\theta_t)\|^2 + \frac{5}{8}\eta_\theta \left[\beta_{\theta w}^2 \mathbb{E}\|w^*(\theta_t) - w_t\|^2 + d_\theta \frac{\sigma_\theta^2}{N} + \frac{4L_\theta^2}{m}\mathbb{1}_{\{m<\tilde{n}\}}\right].$$

In the second inequality, we simply used the definition of $\eta_\theta$. In the third inequality, we used the fact that $\nabla\Phi(\theta_t) = \nabla_\theta F(\theta_t, w^*(\theta_t))$ (see Lemma 15) together with Assumption 13 (part 3). $\qquad\square$

**Lemma 20.** *Grant Assumption 13. If $\eta_w = \frac{1}{\beta_w}$, then the iterates of Algorithm 3 satisfy ($\forall t \geqslant 0$)*

$$\mathbb{E}\|w^*(\theta_{t+1}) - w_{t+1}\|^2 \leqslant \left(1 - \frac{1}{2\kappa_w} + 4\kappa_w \kappa_{\theta w}^2 \eta_\theta^2 \beta_{\theta w}^2\right)\mathbb{E}\|w^*(\theta_t) - w_t\|^2 + \frac{2}{\beta_w^2}\left(\frac{4L_w^2}{m}\mathbb{1}_{\{m<\tilde{n}\}} + d_w \frac{\sigma_w^2}{N}\right)$$

$$+ 4\kappa_w \kappa_{\theta w}^2 \eta_\theta^2 \left(\mathbb{E}\|\nabla\Phi(\theta_t)\|^2 + d_\theta \frac{\sigma_\theta^2}{N}\right).$$

*Proof.* Fix any t and denote $\tilde{g}_w := \frac{1}{N}\sum_{i=1}^N \left(\frac{1}{m}\sum_{i=1}^m \nabla_w f(\theta_{t-1}, w_{t-1}; z_{ji}) + v_{t-1,i}\right) := g_w + v_{t-1}$, so $w_t = w_{t-1} - \eta_\theta \tilde{g}_w$, where $v_{t-1} = \frac{1}{N}\sum_{i=1}^N v_{t-1,i}$. Since $v_{t-1,1}, v_{t-1,2}, ..., v_{t-1,N}$ are i.i.d. sampled from $\mathcal{N}(0, \sigma_w^2)$, we have that $v_{t-1} \sim \mathcal{N}(0, \frac{\sigma_w^2}{N})$. Now take $\delta_t := \mathbb{E}\|w^*(\theta_t) - w_t\|^2 := \mathbb{E}\|w^* - w_t\|^2$. We may assume without loss of generality that $f(\theta, \cdot; z)$ is $\mu$-strongly *convex* and that $w_{t+1} = \Pi_\mathcal{W}[w_t - \frac{1}{\beta_w}\tilde{g}_w] := \Pi_\mathcal{W}[w_t - \frac{1}{\beta_w}(g_w + v_t)]$. Now,

$$\mathbb{E}\|w_{t+1} - w^*\|^2 = \mathbb{E}\left\|\Pi_\mathcal{W}[w_t - \frac{1}{\beta_w}\tilde{g}_w] - w^*\right\|^2 \leqslant \mathbb{E}\left\|w_t - \frac{1}{\beta_w}\tilde{g}_w - w^*\right\|^2$$

$$= \mathbb{E}\|w_t - w^*\|^2 + \frac{1}{\beta_w^2}\left[\mathbb{E}\|g_w\|^2 + d_w \frac{\sigma_w^2}{N}\right] - \frac{2}{\beta_w}\mathbb{E}\langle w_t - w^*, \tilde{g}_w\rangle$$

$$\leqslant \mathbb{E}\|w_t - w^*\|^2 + \frac{1}{\beta_w^2}\left[\mathbb{E}\|g_w\|^2 + d_w \frac{\sigma_w^2}{N}\right] - \frac{2}{\beta_w}\mathbb{E}\left[F(\theta_t, w_t) - F(\theta_t, w^*) + \frac{\mu}{2}\|w_t - w^*\|^2\right]$$

$$\leqslant \delta_t\left(1 - \frac{\mu}{\beta_w}\right) - \frac{2}{\beta_w}\mathbb{E}\left[F(\theta_t, w_t) - F(\theta_t, w^*)\right] + \frac{\mathbb{E}\|g_w\|^2}{\beta_w^2} + \frac{d_w \sigma_w^2}{N\beta_w^2}.$$

Rewriting,

$$\mathbb{E}\|g_w\|^2 = \mathbb{E}\left[\|g_w - \nabla_w F(\theta_t, w_t)\|^2 + \|\nabla_w F(\theta_t, w_t)\|^2\right]$$

$$\leqslant \mathbb{E}\left[\|g_w - \nabla_w F(\theta_t, w_t)\|^2\right] + 2\beta_w[F(\theta_t, w_t) - F(\theta_t, w^*(\theta_t))],$$

using $\mathbb{E}g_w = \nabla_w F(\theta_t, w_t)$ in the first equality, and Lemma 17 (plus Assumption 13 part 5) in the second inequality.

Expanding, $\mathbb{E}\left[\|g_w - \nabla_w F(\theta_t, w_t)\|^2\right]$ in the same manner as we did for $\mathbb{E}\left[\|g - \nabla_\theta F(\theta_t, w_t)\|^2\right]$ in Lemma 18, we get that

$$\mathbb{E}\left[\|g_w - \nabla_w F(\theta_t, w_t)\|^2\right] \leqslant \frac{4\mathbb{1}_{\{m<\tilde{n}\}}L_w^2}{m}$$

This implies

$$\mathbb{E}\|w_{t+1} - w^*\|^2 \leqslant \delta_t\left(1 - \frac{1}{\kappa_w}\right) + \frac{1}{\beta_w^2}\left[d_w \frac{\sigma_w^2}{N} + \frac{4L_w^2}{m}\mathbb{1}_{\{m<\tilde{n}\}}\right]. \tag{9}$$

Therefore,

$$
\begin{aligned}
\delta_{t+1} &= \mathbb{E}\|w_{t+1} - w^*(\theta_t) + w^*(\theta_t) - w^*(\theta_{t+1})\|^2 \\
&\leqslant \left(1 + \frac{1}{2\kappa_w - 1}\right)\mathbb{E}\|w_{t+1} - w^*(\theta_t)\|^2 + 2\kappa_w\mathbb{E}\|w^*(\theta_t) - w^*(\theta_{t+1})\|^2 \\
&\leqslant \left(1 + \frac{1}{2\kappa_w - 1}\right)\left(1 - \frac{1}{\kappa_w}\right)\delta_t + \frac{2}{\beta_w^2}\left[d_w\frac{\sigma_w^2}{N} + \frac{4L_w^2}{m}\mathbb{1}_{\{m < \tilde{n}\}}\right] + 2\kappa_w\mathbb{E}\|w^*(\theta_t) - w^*(\theta_{t+1})\|^2 \\
&\leqslant \left(1 + \frac{1}{2\kappa_w - 1}\right)\left(1 - \frac{1}{\kappa_w}\right)\delta_t + \frac{2}{\beta_w^2}\left[d_w\frac{\sigma_w^2}{N} + \frac{4L_w^2}{m}\mathbb{1}_{\{m < \tilde{n}\}}\right] + 2\kappa_w\kappa_{\theta w}^2\mathbb{E}\|\theta_t - \theta_{t+1}\|^2 \\
&\leqslant \left(1 + \frac{1}{2\kappa_w - 1}\right)\left(1 - \frac{1}{\kappa_w}\right)\delta_t + \frac{2}{\beta_w^2}\left[d_w\frac{\sigma_w^2}{N} + \frac{4L_w^2}{m}\mathbb{1}_{\{m < \tilde{n}\}}\right] \\
&\quad + 4\kappa_w\kappa_{\theta w}^2\eta_\theta^2\left[\mathbb{E}\|\nabla_\theta F(\theta_t, w_t) - \nabla\Phi(\theta_t)\|^2 + \|\nabla\Phi(\theta_t)\|^2 + d_\theta\frac{\sigma_\theta^2}{N}\right] \\
&= \left(1 + \frac{1}{2\kappa_w - 1}\right)\left(1 - \frac{1}{\kappa_w}\right)\delta_t + \frac{2}{\beta_w^2}\left[d_w\frac{\sigma_w^2}{N} + \frac{4L_w^2}{m}\mathbb{1}_{\{m < \tilde{n}\}}\right] \\
&\quad + 4\kappa_w\kappa_{\theta w}^2\eta_\theta^2\left[\mathbb{E}\|\nabla_\theta F(\theta_t, w_t) - \nabla_\theta F(\theta_t, w^*(\theta_t))\|^2 + \|\nabla\Phi(\theta_t)\|^2 + d_\theta\frac{\sigma_\theta^2}{N}\right] \\
&\leqslant \left(1 + \frac{1}{2\kappa_w - 1}\right)\left(1 - \frac{1}{\kappa_w}\right)\delta_t + \frac{2}{\beta_w^2}\left[d_w\frac{\sigma_w^2}{N} + \frac{4L_w^2}{m}\mathbb{1}_{\{m < \tilde{n}\}}\right] \\
&\quad + 4\kappa_w\kappa_{\theta w}^2\eta_\theta^2\left[\beta_{\theta w}^2\mathbb{E}\|w_t - w^*(\theta_t)\|^2 + \|\nabla\Phi(\theta_t)\|^2 + d_\theta\frac{\sigma_\theta^2}{N}\right],
\end{aligned}
$$

by Young's inequality, (9), and Lemma 15. Since $\left(1 + \frac{1}{2\kappa_w - 1}\right)\left(1 - \frac{1}{\kappa_w}\right) \leqslant 1 - \frac{1}{2\kappa_w}$, we obtain

$$
\delta_{t+1} \leqslant \left(1 - \frac{1}{2\kappa_w} + 4\kappa_w\kappa_{\theta w}^2\eta_\theta^2\beta_{\theta w}^2\right)\delta_t + \frac{2}{\beta_w^2}\left[d_w\frac{\sigma_w^2}{N} + \frac{4L_w^2}{m}\mathbb{1}_{\{m < \tilde{n}\}}\right] + 4\kappa_w\kappa_{\theta w}^2\eta_\theta^2\left[\|\nabla\Phi(\theta_t)\|^2 + d_\theta\frac{\sigma_\theta^2}{N}\right],
$$

as desired. $\qquad\square$

*Proof.* (of Theorem 14) **Privacy:** By the definition of ISRL-DP, independence of the Gaussian noise across silos, and symmetry of Algorithm 3 w.r.t. every silo, it suffices to show that the full transcript of silo $j$'s communications is $(\varepsilon, \delta)$-DP for any fixed settings of other silos' messages and data. But, the prescribed choices of noise in Theorem 14 follows directly from the calculations of Theorem E.1 Lowy et al. (2023) *with respect to a single silo*. Hence, the privacy of the communication transcript of data follows directly from the analysis of Lowy et al. (2023). Therefore, Algorithm 3 is $(\varepsilon, \delta)$-ISRL-DP.

**Convergence:**

Denote $\zeta := 1 - \frac{1}{2\kappa_w} + 4\kappa_w\kappa_{\theta w}^2\eta_\theta^2\beta_{\theta w}^2$, $\delta_t = \mathbb{E}\|w^*(\theta_t) - w_t\|^2$, and

$$
C_t := \frac{2}{\beta_w^2}\left(\frac{4L_w^2}{m}\mathbb{1}_{\{m < \tilde{n}\}} + d_w\frac{\sigma_w^2}{N}\right) + 4\kappa_w\kappa_{\theta w}^2\eta_\theta^2\left(\mathbb{E}\|\nabla\Phi(\theta_t)\|^2 + d_\theta\frac{\sigma_\theta^2}{N}\right),
$$

so that Lemma 20 reads as

$$
\delta_t \leqslant \zeta\delta_{t-1} + C_{t-1} \tag{10}
$$

for all $t \in [T]$. Applying (10) recursively, we have

$$\delta_t \leqslant \zeta^t \delta_0 + \sum_{j=0}^{t-1} C_{t-j-1} \zeta^j$$

$$\leqslant \zeta^t D^2 + 4\kappa_w \kappa_{\theta w}^2 \eta_\theta^2 \sum_{j=0}^{t-1} \zeta^{t-1-j} \mathbb{E}\|\nabla\Phi(\theta_j)\|^2$$

$$+ \left(\sum_{j=0}^{t-1} \zeta^{t-1-j}\right) \left[\frac{2}{\beta_w^2}\left(\frac{4L_w^2}{m}\mathbb{1}_{\{m<\tilde{n}\}} + d_w\frac{\sigma_w^2}{N}\right) + 4\kappa_w \kappa_{\theta w}^2 \eta_\theta^2 d_\theta \frac{\sigma_\theta^2}{N}\right].$$

Combining this inequality with Lemma 19, we get

$$\mathbb{E}\Phi(\theta_t) \leqslant \mathbb{E}\left[\Phi(\theta_{t-1}) - \frac{3}{8}\eta_\theta\|\nabla\Phi(\theta_{t-1})\|^2\right] + \frac{5}{8}\eta_\theta\left(d_\theta\frac{\sigma_\theta^2}{N} + \frac{4L_\theta^2}{m}\mathbb{1}_{\{m<\tilde{n}\}}\right)$$

$$+ \frac{5}{8}\eta_\theta\beta_{\theta w}^2\left\{\zeta^t D^2 + 4\kappa_w\kappa_{\theta w}^2\eta_\theta^2 \sum_{j=0}^{t-1} \zeta^{t-1-j}\mathbb{E}\|\nabla\Phi(\theta_j)\|^2\right.$$

$$\left. + \left(\sum_{j=0}^{t-1}\zeta^{t-1-j}\right)\left[\frac{2}{\beta_w^2}\left(\frac{4L_w^2}{m}\mathbb{1}_{\{m<\tilde{n}\}} + d_w\frac{\sigma_w^2}{N}\right) + 4\kappa_w\kappa_{\theta w}^2\eta_\theta^2 d_\theta\frac{\sigma_\theta^2}{N}\right]\right\}.$$

Summing over all $t \in [T]$ and re-arranging terms yields

$$\mathbb{E}\Phi(\theta_T) \leqslant \Phi(\theta_0) - \frac{3}{8}\eta_\theta\sum_{t=1}^T \mathbb{E}\|\nabla\Phi(\theta_{t-1})\|^2 + \frac{5}{8}\eta_\theta T\left(d_\theta\frac{\sigma_\theta^2}{N} + \frac{4L_\theta^2}{m}\mathbb{1}_{\{m<\tilde{n}\}}\right) + \frac{5}{8}\eta_\theta\beta_{\theta w}^2\left(\sum_{t=1}^T \zeta^t\right)D^2$$

$$+ 4\eta_\theta^3\beta_{\theta w}^2\kappa_w\kappa_{\theta w}^2\sum_{t=1}^T\sum_{j=0}^{t-1}\zeta^{t-1-j}\mathbb{E}\|\nabla\Phi(\theta_j)\|^2$$

$$+ \frac{5}{8}\left(\sum_{t=1}^T\sum_{j=0}^{t-1}\zeta^{t-1-j}\right)\eta_\theta\beta_{\theta w}^2\left[\frac{2}{\beta_w^2}\left(\frac{4L_w^2}{m}\mathbb{1}_{\{m<\tilde{n}\}} + d_w\frac{\sigma_w^2}{N}\right) + 4\kappa_w\kappa_{\theta w}^2\eta_\theta^2 d_\theta\frac{\sigma_\theta^2}{N}\right].$$

Now, $\zeta \leqslant 1 - \frac{1}{4\kappa_w}$, which implies that

$$\sum_{t=1}^T \zeta^t \leqslant 4\kappa_w \quad \text{and}$$

$$\sum_{t=1}^T\sum_{j=0}^{t-1}\zeta^{t-1-j} \leqslant 4\kappa_w T.$$

Hence

$$\frac{1}{T}\sum_{t=1}^T\mathbb{E}\|\nabla\Phi(\theta_t)\|^2 \leqslant \frac{3[\Phi(\theta_0) - \mathbb{E}\Phi(\theta_T)]}{\eta_\theta T} + \frac{5}{3}\left(d_\theta\frac{\sigma_\theta^2}{N} + \frac{4L_\theta^2}{m}\mathbb{1}_{\{m<\tilde{n}\}}\right) + \frac{7\beta_{\theta w}^2 D^2\kappa_w}{T}$$

$$+ \frac{48\eta_\theta^2\beta_{\theta w}^2\kappa_w^2\kappa_{\theta w}^2}{T}\left(\sum_{t=1}^T\mathbb{E}\|\nabla\Phi(\theta_t)\|^2\right)$$

$$+ 8\kappa_w\beta_{\theta w}^2\frac{2}{\beta_w^2}\left(\frac{4L_w^2}{m}\mathbb{1}_{\{m<\tilde{n}\}} + d_w\frac{\sigma_w^2}{N}\right) + 32\beta_{\theta w}^2\kappa_w^2\kappa_{\theta w}^2\eta_\theta^2 d_\theta\frac{\sigma_\theta^2}{N}.$$

Since $\eta_\theta^2\beta_{\theta w}^2\kappa_w^2\kappa_{\theta w}^2 \leqslant \frac{1}{256}$, we obtain

$$\mathbb{E}\|\nabla\Phi(\hat{\theta}_T)\|^2 \lesssim \frac{\Delta_\Phi\kappa_w}{T}(\beta_\theta + \beta_{\theta w}\kappa_{\theta w}) + \frac{d_\theta L_\theta^2 T\ln(1/\delta)}{\varepsilon^2\tilde{n}^2 N} + \frac{1}{m}\mathbb{1}_{\{m<\tilde{n}\}}\left(L_\theta^2 + \frac{\kappa_w\beta_{\theta w}^2 L_w^2}{\beta_w^2}\right) + \frac{\kappa_w\beta_{\theta w}^2 L_w^2 d_w T\ln(1/\delta)}{\beta_w^2\varepsilon^2\tilde{n}^2 N}$$

$$+ \frac{\beta_{\theta w}^2 D^2\kappa_w}{T}.$$

Our choice of $T$ then implies

$$\mathbb{E}\|\nabla\Phi(\hat{\theta}_T)\|^2 \lesssim \sqrt{\Delta_\Phi\left(\beta_\theta + \beta_{\theta w}\kappa_{\theta w}\right)\kappa_w + \kappa_w\beta_{\theta w}^2 D^2}\left[\frac{L_\theta\sqrt{d_\theta\ln(1/\delta)}}{\varepsilon\tilde{n}\sqrt{N}} + \left(\frac{\beta_{\theta w}\sqrt{\kappa_w}}{\beta_w}\right)\frac{L_w\sqrt{d_w\ln(1/\delta)}}{\varepsilon\tilde{n}\sqrt{N}}\right]$$
$$+ \frac{\mathbb{1}_{\{m<n\}}}{m}\left(L_\theta^2 + \frac{\kappa_w\beta_{\theta w}^2 L_w^2}{\beta_w^2}\right).$$

Finally, our choice of sufficiently large $m$ yields the last claim in Theorem 14. $\qquad\square$

### D.2 Proof of Theorem 7: Utility Claim

The utility claim in Theorem 7 is an easy consequence of Theorem 14, which we proved above:

**Theorem 21** (Precise Re-statement of Utility Claim in Theorem 7)**.** *Assume the loss function $\ell(\cdot, x, y)$ and $\mathcal{F}(x, \cdot)$ are Lipschitz continuous with Lipschitz gradient for all $(x, y)$, and $\widehat{P}_S(r) \geqslant \rho > 0 \ \forall \ r \in [k]$. In Algorithm 1, choose $\mathcal{W}$ to be a sufficiently large ball that contains $W^*(\theta) := \mathrm{argmax}_W \widehat{F}(\theta, W)$ for every $\theta$ in some neighborhood of $\theta^* \in \mathrm{argmin}_\theta \max_W \widehat{F}(\theta, W)$. Then there exist algorithmic parameters such that the $(\varepsilon, \delta)$-ISRL-DP Algorithm 1 returns $\hat{\theta}_T$ with*

$$\mathbb{E}\|\nabla FERMI(\hat{\theta}_T)\|^2 = \mathcal{O}\left(\frac{\sqrt{\max(d_\theta, kl)\ln(1/\delta)}}{\varepsilon\tilde{n}\sqrt{N}}\right),$$

*treating $D = diameter(\mathcal{W})$, $\lambda$, $\rho$, and the Lipschitz and smoothness parameters of $\ell$ and $\mathcal{F}$ as constants.*

*Proof.* By Theorem 14, it suffices to show that Assumption 13 holds for $f(\theta, W; z_i) := \ell(\theta, x_i, y_i) + \lambda\widehat{\psi}_i(\theta, W)$. We assumed $\ell(\cdot, x_i, y_i)$ is Lipschitz continuous with Lipschitz gradient. Further, the work of Lowy et al. (2022a) showed that $f(\theta, \cdot; z_i)$ is strongly concave. Thus, it suffices to show that $\widehat{\psi}_i(\theta, W)$ is Lipschitz continuous with Lipschitz gradient. This directly holds from Lemma 9 as $\mathcal{F}$ is lipschitz continuous and $\mathcal{W}$ has bounded diameter. $\qquad\square$

## E  Numerical Experiments: Additional Details and Results

The code for this work can be found at https://anonymous.4open.science/r/Stochastic-Federated-Differentially-Private-and-Fair-Learning-0DB7.

### E.1  Measuring Demographic Parity and Equalized Odds Violation

Following the evaluation setup from Lowy et al. (2023). We used the expressions given in (11) and (12) to measure the demographic parity violation and the equalized odds violation respectively. We denote $\mathcal{Y}$ to be the set of all possible output classes and $\mathcal{S}$ to be the classes of the sensitive attribute. $P[E]$ denotes the empirical probability of the occurrence of an event E. We subsequently present the results for the Demographic Parity and misclassification error trade-offs for the Adult and Retired Adult Datasets and results for Equalized odds on the credit card dataset.

$$\max_{y'\in\mathcal{Y}, s_1, s_2\in\mathcal{S}}\left|P[\hat{y} = y'|s = s_1] - P[\hat{y} = y'|s = s_2]\right| \tag{11}$$

$$\max_{y'\in\mathcal{Y}, s_1, s_2\in\mathcal{S}}\max(\left|P[\hat{y} = y'|s = s_1, y = y'] - P[\hat{y} = y'|s = s_2, y = y']\right|,$$
$$\left|P[\hat{y} = y'|s = s_1, y \neq y'] - P[\hat{y} = y'|s = s_2, y \neq y']\right|) \tag{12}$$

### E.2 Tabular Datasets

#### E.2.1 Model Description and Experimental Details

**Demographic Parity:** We split each dataset in a 3:1 train:test ratio. We preprocess the data similar to Hardt et al. (2016a) and use a simple logistic regression model with a sigmoid output $O = \sigma(Wx + b)$ which we treat as conditional probabilities $p(\hat{y} = i|x)$. The predicted variables and sensitive attributes are both binary in this case across all the datasets. We analyze fairness-accuracy trade-offs with three different values of $\varepsilon \in \{1, 3, 9\}$ for each dataset. We compare against state-of-the-art algorithms proposed in Ling et al. (2024) and (the demographic parity objective of) Tran et al. (2021). The tradeoff curves for SteFFLe were generated by sweeping across different values for $\lambda \in [0, 2.0]$. The learning rates for the descent and ascent, $\eta_\theta$ and $\eta_w$ of which the former was subjected to a step decay of 0.8 every 10 epochs while the latter was kept constant during the optimization process with the initial values at 0.25 and $1e - 5$, respectively. Batch size was 256. We tuned the $\ell_2$ diameter of the projection set $\mathcal{W}$ and $\theta$-gradient clipping threshold in $[1, 5]$ in order to generate stable results with high privacy (i.e. low $\varepsilon$). Each model was trained for 40 epochs. The results displayed are averages over 15 trials (random seeds) for each value of $\varepsilon$ and heterogeneity level $h$.

**Equalized Odds:** We replicated the experimental setup described above, but we took $\ell_2$ diameter of $\mathcal{W}$ and the value of gradient clipping for $\theta$ to be in $[1, 2]$. Also, we only tested two values of $\varepsilon \in \{1, 3\}$ and two values of heterogeneity levels $h \in \{0, 0.75\}$.

#### E.2.2 Description of Datasets

**Adult Income Dataset:** This dataset contains the census information about the individuals. The classification task is to predict whether the person earns more than 50k every year or not. We followed a preprocessing approach similar to Lowy et al. (2022a). After preprocessing, there were a total of 102 input features from this dataset. The sensitive attribute for this work in this dataset was taken to be gender. This dataset consists of around 48,000 entries spanning across two CSV files, which we combine and then we take the train-test split of 3:1.

**Retired Adult Income Dataset:** The Retired Adult Income Dataset proposed by Ding et al. (2021) is essentially a superset of the Adult Income Dataset which attempts to counter some caveats of the Adult dataset. The input and the output attributes for this dataset is the same as that of the Adult Dataset and the sensitive attribute considered in this work is the same as that of the Adult. This dataset contains around 45,000 entries.

**Credit Card Dataset:** This dataset contains the financial data of users in a bank in Taiwan consisting of their gender, education level, age, marital status, previous bills, and payments. The assigned classification task is to predict whether the person defaults their credit card bills or not, essentially making the task if the clients were credible or not. We followed a preprocessing approach similar to Lowy et al. (2022a). After preprocessing, there were a total of 85 input features from this dataset. The sensitive attribute for this dataset was taken to be gender. This dataset consists of around 30,000 entries from which we take the train-test split of 3:1.

### E.3 Large scale Image Datasets

#### E.3.1 Model Description and Experimental Details

We trained a ResNet-18 classifier (He et al., 2016) with $d \approx 11.68$ million parameters on the UTK-Face dataset (Yang & Hairong, 2017) using 1. We classified facial images into 9 age groups, following Tran et al. (2022), with *race* (5 classes) as the sensitive attribute to measure demographic parity violations. Each model was trained for 250 epochs, with results from a randomly selected epoch averaged across 7 runs, each with a different random seed per dataset and privacy setting. To generate tradeoff curves, we varied the fairness regularization parameter $\lambda$ by sweeping it over the range $[0, 500]$. The dataset was split 3:1 into training and testing sets. Images were resized and normalized to $128 \times 128$ tensors, with a batch size of 128. We

---

**Algorithm 4** Modelling Heterogeneity Levels in Silo Data

---

1: **Input:** Set of data points $D$, number of silos $K$, total points $n$, attribute for partitioning $A$, level of heterogeneity $h$
2: **Output:** Partitioned subsets $\{D_1, D_2, \ldots, D_K\}$
3: Divide data into equal sized partitions indexed $L \in \{0, 1, \ldots, K-1\}^n$ based on attribute $A$.
4: Initialize vector $P$ of length $n$, representing initial partition assignments.
5: Initialize list $S = []$, to store indices for each silo.
6: **for** each silo $k$ from 0 to $K-1$ **do**
7:     Let $I_k = \{i \mid P[i] = k\}$, indices where partition $P$ is $k$.
8:     Sample $m_k = \lfloor \frac{n}{K} \times h \rfloor$ unique indices from $I_k$ without replacement.
9:     Set $P[m_k]$ to an unused label (e.g., $K$), indicating these indices are selected.
10:     Add $m_k$ to list $S_k$ in $S$.
11: **end for**
12: **for** each silo $k$ from 0 to $K-1$ **do**
13:     Let $C = \{i \mid P[i] \neq K\}$, indices not yet assigned.
14:     Calculate remaining indices needed: $r_k = \frac{n}{K} - |S_k|$.
15:     Sample $r_k$ indices from $C$ without replacement.
16:     Update $P$ for these indices to $K$, marking them as selected.
17:     Add these indices to the respective list $S_k$ in $S$.
18: **end for**
19: Update $P$ such that all indices in each $S_k$ are set to $k$, forming final partitions.
20: Split $D$ into $\{D_1, D_2, \ldots, D_K\}$ based on updated partitions $P$.
21: **return** $\{D_1, D_2, \ldots, D_K\}$

---

regulated optimization by adjusting the $\ell_2$ diameter of $\mathcal{W}$ and clipping gradients of $\theta$ within $[1, 2]$. The ascent and descent learning rates, $\eta_\theta$ and $\eta_w$, were fixed at 0.001 and 0.0005.

To balance fairness and accuracy, we experimented with privacy parameters $\epsilon \in \{10, 25, 50\}$ under homogeneous ($h = 0$) and heterogeneous ($h = 0.75$) settings. Due to the large $d$, stable results required higher $\epsilon$, as privacy cost scales with $d$ (see Theorem 21). In contrast, smaller-scale experiments in (Jagielski et al., 2019) used $\epsilon > 100$. We used (Tran et al., 2021; Ling et al., 2024) as baselines, adapting them for large-scale, non-binary classification. However, (Tran et al., 2021) was highly unstable, often predicting the same class regardless of input, so we excluded its results. Our method remained stable, producing meaningful tradeoff curves. Although (Tran et al., 2022) reported results on UTK-Face, their code was unavailable, preventing replication.

### E.3.2 Description of Datasets

**UTK-Face Dataset:** This large-scale image dataset, introduced by Yang & Hairong (2017), includes facial images spanning ages from 0 to 116. Comprising over 20,000 images, it provides annotations for age, gender, and ethnicity while encompassing significant variations in facial expressions, poses, lighting conditions, occlusions, and resolutions. For our experiments, we focus on an age classification task, dividing the age range into 9 groups, following the setup in Tran et al. (2021). Ethnicity, which consists of 5 distinct categories, is treated as the sensitive attribute in our analysis.

### E.4 Modelling Heterogeneity

The proposed algorithm, Algorithm 4, partitions a dataset $D$ of size $n$ into $K$ silos, incorporating a specified heterogeneity level $h$. It begins by dividing the data into equal-sized partitions based on a predefined attribute $A$ (age, in all our experiments), which determines initial partition labels. For each silo $k$, a subset of data points is sampled based on $h$, introducing a controlled deviation from homogeneous partitioning. This ensures that each silo receives approximately $\frac{n}{K}$ data points while allowing for varying levels of overlap across silos. The remaining unassigned points are then distributed to maintain a balance between homogeneous and

heterogeneous data distributions. The final output consists of $K$ subsets $\{D_1, D_2, \ldots, D_K\}$, with $h$ governing the diversity of the partitions.

This algorithm offers notable advantages over traditional methods using Dirichlet distributions for modeling heterogeneity in federated learning. In terms of *interpretability*, it provides explicit control over the degree of heterogeneity via the parameter $h$, allowing for easy quantification and adjustment of deviations from homogeneity. In contrast, Dirichlet-based methods abstract this process, making it harder to control or interpret heterogeneity without post-hoc analysis.

In terms of *flexibility*, the algorithm allows granular control over partitioning based on specific attributes and sample sizes, which is not straightforward with Dirichlet-based methods. While Dirichlet distributions often lead to unpredictable or uneven partition sizes, this algorithm ensures each silo receives approximately $\frac{n}{K}$ data points, with controlled deviations through $h$. This flexibility makes it a more practical and customizable solution for federated learning scenarios requiring balanced silo sizes and heterogeneous distributions. Thus, the proposed approach improves both interpretability and flexibility, making it more effective for modeling heterogeneity in federated learning.

### E.5    Additional Numerical Results

### E.5.1    Demographic Parity

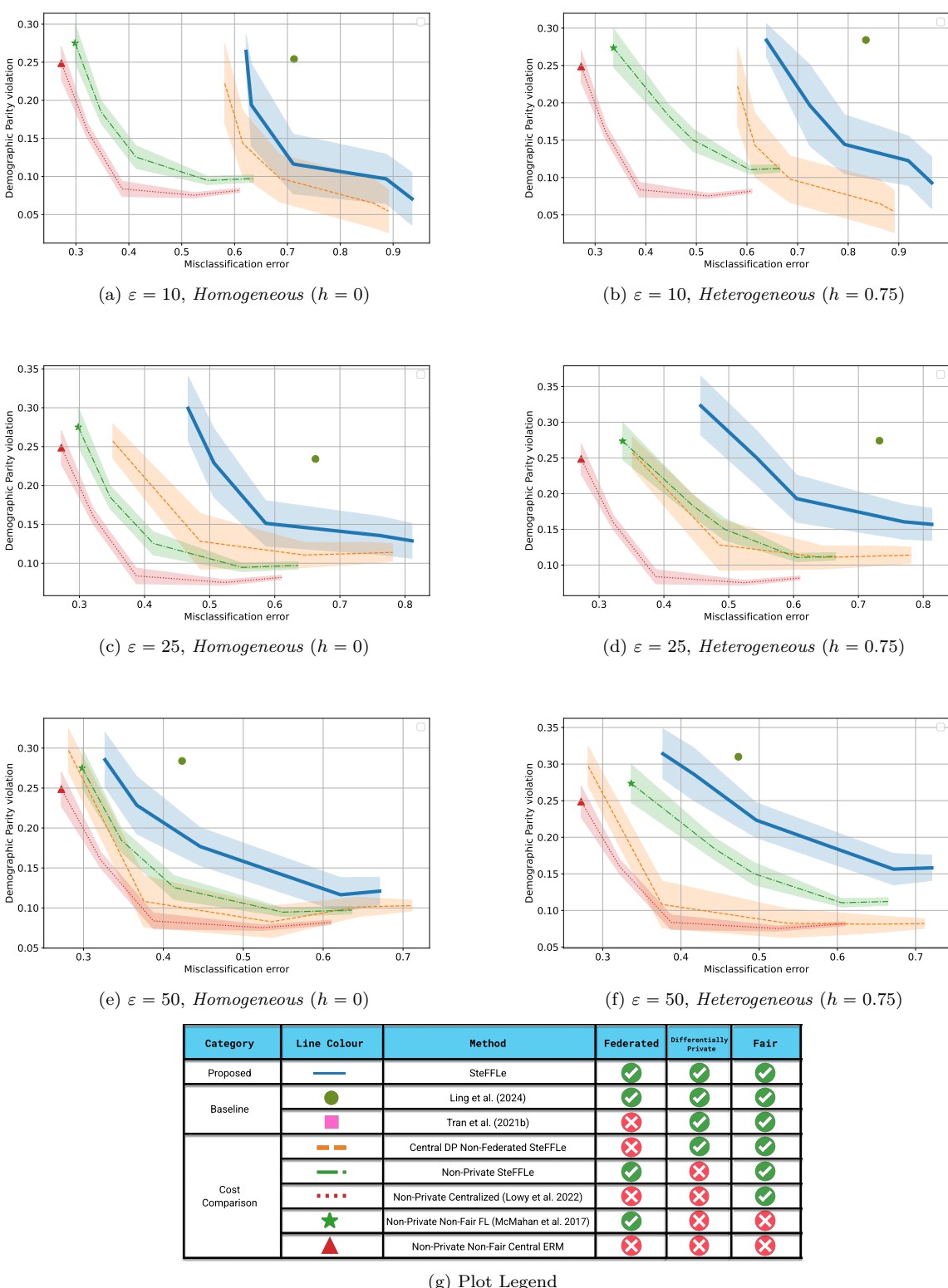

(a) $\varepsilon = 10$, *Homogeneous* ($h = 0$)

(b) $\varepsilon = 10$, *Heterogeneous* ($h = 0.75$)

(c) $\varepsilon = 25$, *Homogeneous* ($h = 0$)

(d) $\varepsilon = 25$, *Heterogeneous* ($h = 0.75$)

(e) $\varepsilon = 50$, *Homogeneous* ($h = 0$)

(f) $\varepsilon = 50$, *Heterogeneous* ($h = 0.75$)

| Category | Line Colour | Method | Federated | Differentially Private | Fair |
|----------|-------------|--------|-----------|------------------------|------|
| Proposed | —— | SteFFLe | ✓ | ✓ | ✓ |
| Baseline | ● | Ling et al. (2024) | ✓ | ✓ | ✓ |
| | ■ | Tran et al. (2021b) | ✗ | ✓ | ✓ |
| Cost Comparison | – – – | Central DP Non-Federated SteFFLe | ✗ | ✓ | ✓ |
| | – · – · | Non-Private SteFFLe | ✓ | ✗ | ✓ |
| | · · · · | Non-Private Centralized (Lowy et al. 2022) | ✗ | ✗ | ✓ |
| | ★ | Non-Private Non-Fair FL (McMahan et al. 2017) | ✓ | ✗ | ✗ |
| | ▲ | Non-Private Non-Fair Central ERM | ✗ | ✗ | ✗ |

(g) Plot Legend

Figure 3: Demographic parity vs Misclassification error on the *UTK-Face image* dataset (*Number of Silos = 3*)

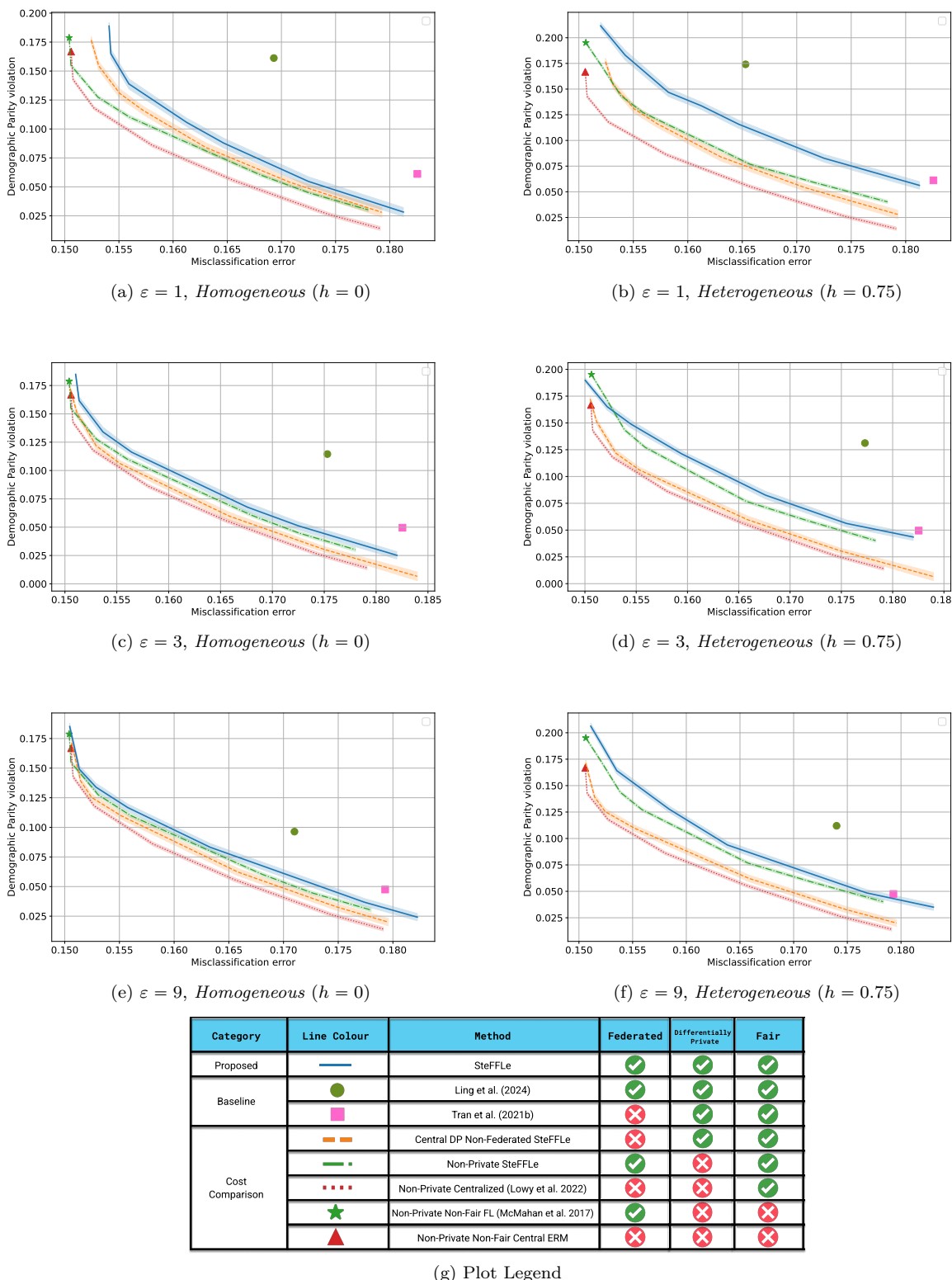

(a) $\varepsilon = 1$, *Homogeneous* ($h = 0$)

(b) $\varepsilon = 1$, *Heterogeneous* ($h = 0.75$)

(c) $\varepsilon = 3$, *Homogeneous* ($h = 0$)

(d) $\varepsilon = 3$, *Heterogeneous* ($h = 0.75$)

(e) $\varepsilon = 9$, *Homogeneous* ($h = 0$)

(f) $\varepsilon = 9$, *Heterogeneous* ($h = 0.75$)

(g) Plot Legend

Figure 4: Demographic parity vs Misclassification error on *Adult* dataset (*Number of Silos* = 3)

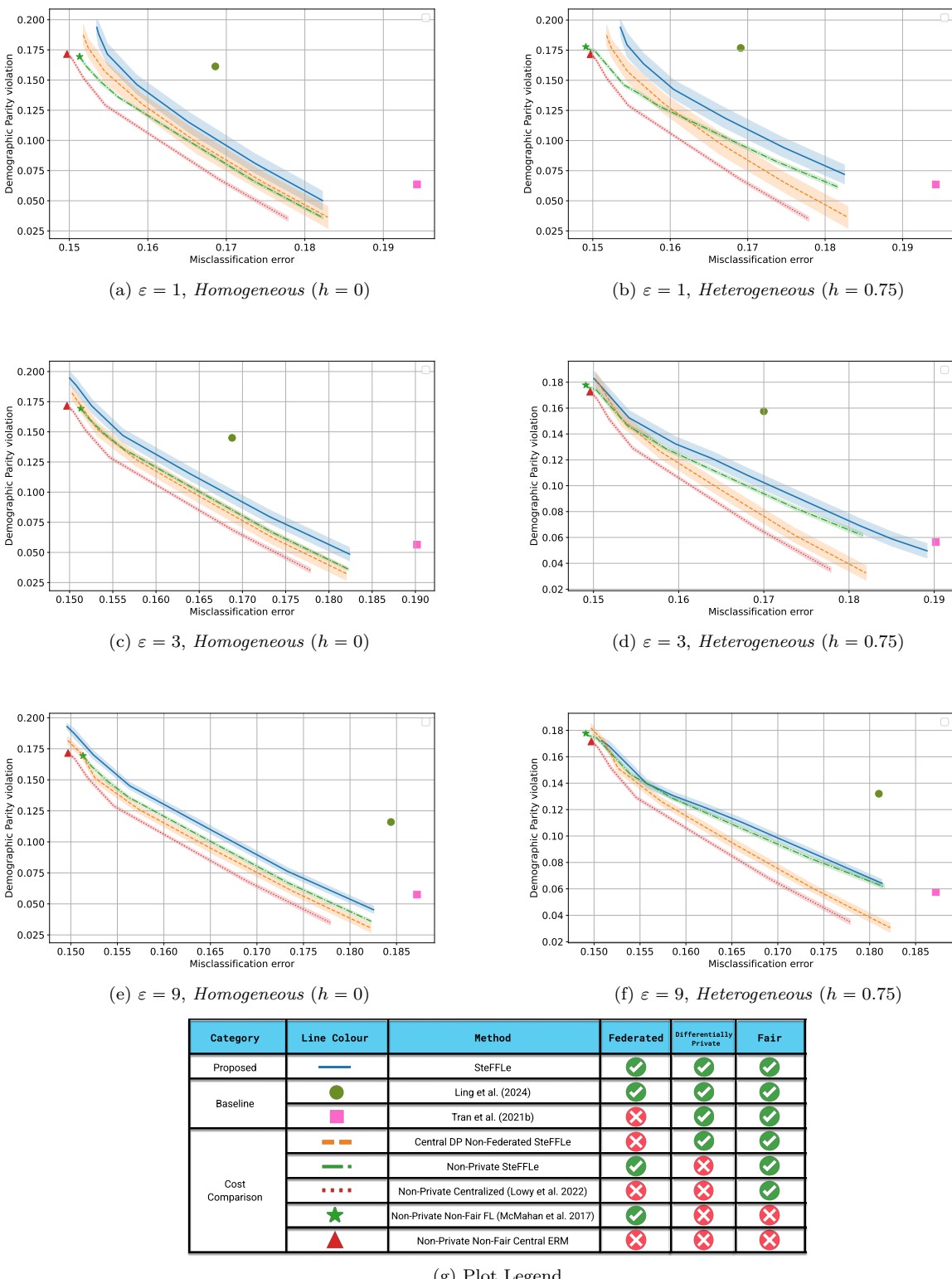

(a) $\varepsilon = 1$, *Homogeneous* ($h = 0$)

(b) $\varepsilon = 1$, *Heterogeneous* ($h = 0.75$)

(c) $\varepsilon = 3$, *Homogeneous* ($h = 0$)

(d) $\varepsilon = 3$, *Heterogeneous* ($h = 0.75$)

(e) $\varepsilon = 9$, *Homogeneous* ($h = 0$)

(f) $\varepsilon = 9$, *Heterogeneous* ($h = 0.75$)

(g) Plot Legend

Figure 5: Demographic parity vs Misclassification error on *Retired Adult* dataset (*Number of Silos* = 3)

### E.5.2 Equalized Odds

In this section, we now focus on a modified version of Algorithm 1, which aims to minimize the violation of Equalized Odds. This is achieved by replacing the absolute probabilities in the objective function with class-conditional probabilities, as detailed in Equation 12.

For this set of experiments, we used the Credit Card dataset, keeping the sensitive attributes and output labels consistent with the previous sections. The results specific to the Credit Card dataset can be found in Appendix E.5.2. When compared to the methods proposed by Ling et al. (2024) and the equalized odds objective introduced by Tran et al. (2021), *our Equalized Odds variant of SteFFLe consistently outperforms these state-of-the-art baselines across all privacy and heterogeneity levels*. Notably, our model surpasses the performance of Tran et al. (2021) by an exceptional margin—exceeding their results by over 150%. This significant improvement resulted in a visual distortion of the corresponding plots, leading to their exclusion to preserve the clarity and interpretability of the visual representations.

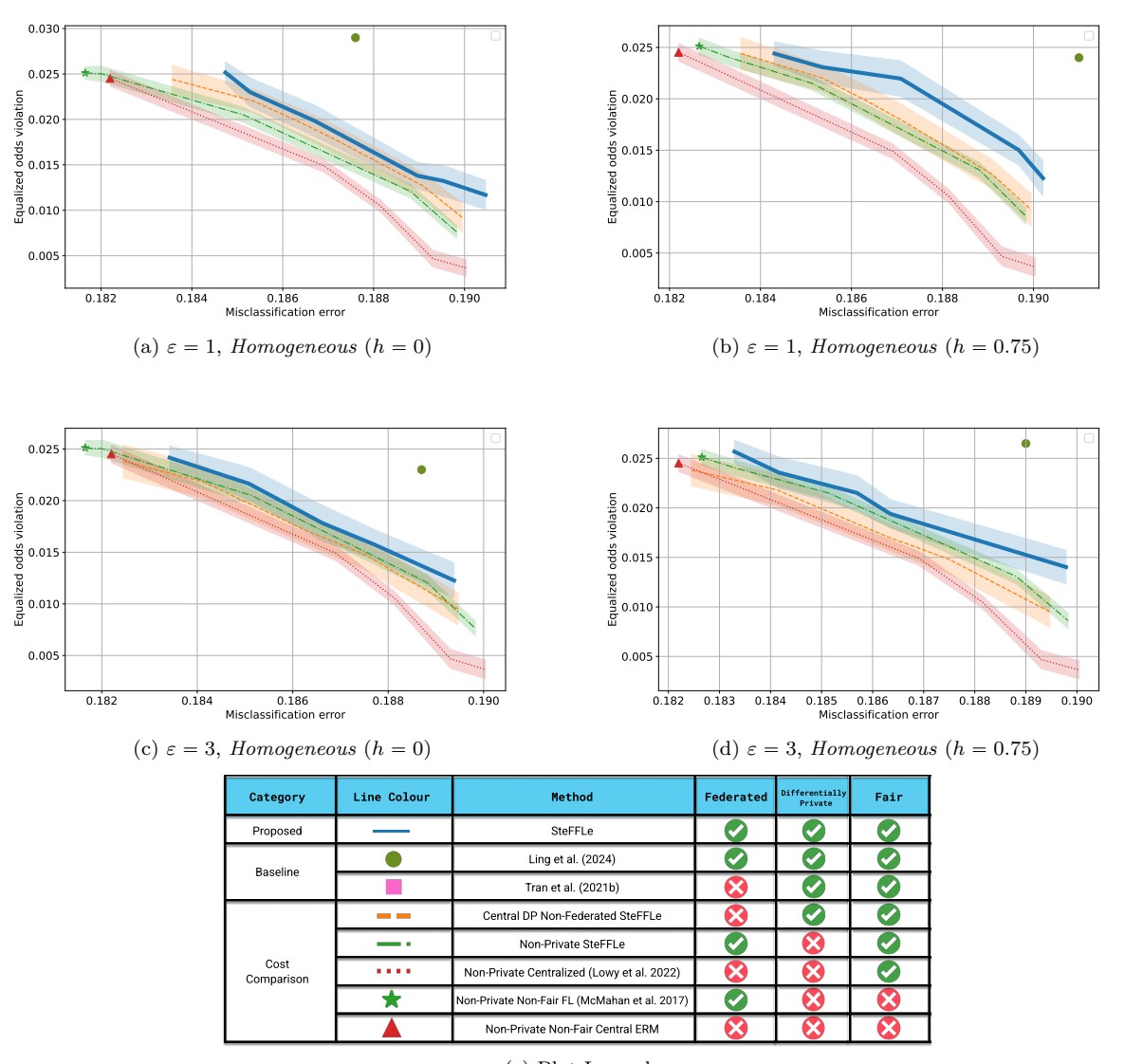

(a) $\varepsilon = 1$, *Homogeneous* ($h = 0$)

(b) $\varepsilon = 1$, *Homogeneous* ($h = 0.75$)

(c) $\varepsilon = 3$, *Homogeneous* ($h = 0$)

(d) $\varepsilon = 3$, *Homogeneous* ($h = 0.75$)

| Category | Line Colour | Method | Federated | Differentially Private | Fair |
|---|---|---|---|---|---|
| Proposed | —— | SteFFLe | ✅ | ✅ | ✅ |
| Baseline | ● | Ling et al. (2024) | ✅ | ✅ | ✅ |
| | ■ | Tran et al. (2021b) | ❌ | ✅ | ✅ |
| Cost Comparison | — — | Central DP Non-Federated SteFFLe | ❌ | ✅ | ✅ |
| | —·— | Non-Private SteFFLe | ✅ | ❌ | ✅ |
| | ···· | Non-Private Centralized (Lowy et al. 2022) | ❌ | ❌ | ✅ |
| | ★ | Non-Private Non-Fair FL (McMahan et al. 2017) | ✅ | ❌ | ❌ |
| | ▲ | Non-Private Non-Fair Central ERM | ❌ | ❌ | ❌ |

(e) Plot Legend

Figure 6: Equalized Odds vs Misclassification error on *Credit Card* dataset (*Number of Silos* = 3)

### E.5.3 Empirical Observations

The experiments with demographic parity and equalized odds (fairness) provide several key insights into the trade-off between misclassification error and fairness across all groups. Across all plots, our model consistently outperforms the baselines proposed by Tran et al. (2021) and Ling et al. (2024), exhibiting substantial performance improvements. Furthermore, as the heterogeneity level increases, the trade-off between fairness and accuracy degrades Fig. 2 and similarly, the trade-off worsens as the number of silos increases Fig. 2. In the equalized odds experiments, *SteFFLe* outperforms both Ling et al. (2024) and especially Tran et al. (2021) by such a wide margin that the scale of the plots becomes distorted, necessitating the exclusion of Tran et al. (2021) from the plots for clarity. Moreover while observing the cost of incorporating federated learning and differential privacy is observed through the gap between the four tradeoff curves. For $\varepsilon = 1$, *FERMI* demonstrates the best trade-off, followed by *non-private SteFFLe*, with *DP-FERMI* and *SteFFLe* trailing behind for both demographic parity and equalized odds objectives. As the privacy budget increases to $\varepsilon = 3$ and $\varepsilon = 9$, we observe that the private, non-federated methods begin to outperform their non-private, federated counterparts, with the other methods following their prior ordering.

In the large-scale image dataset experiments, bandwidths decreased with higher privacy budgets due to reduced noise variance, a trend also clearly observed in the tabular dataset experiments (especially for the Retired Adult dataset). The trade-off follows a similar pattern as tabular datasets: at $\varepsilon = 10$, *FERMI* achieves the best trade-off, followed by *non-private SteFFLe*, with *DP-FERMI* and *SteFFLe* trailing for both demographic parity and equalized odds. As $\varepsilon$ increases to 25 and 50, private, non-federated methods begin to outperform their non-private, federated counterparts, while other trade-offs remain similar. Notably, a clear performance gap arises due to heterogeneity levels, as seen in the homogeneous and heterogeneous plots for the same privacy budget in the federated methods, namely *Non-Private SteFFLe* and *SteFFLe*. Our approach, *SteFFLe*, consistently outperforms Ling et al. (2024), whereas Tran et al. (2021) exhibited degenerate results in all experiments, leading to its exclusion from the plots.

However, this approach may suffer from a minor shortcoming: The attacker may figure out the silos which contain the data by looking at the silos that did not participate. We can curb that by prompting all the silos to broadcast a "dummy" Gaussian noise with zero mean and finite variance. Since the noise has zero mean, the unbiasedness and finite variance of the inner gradient would still be preserved. However, such a technique can slow convergence because of the additional variance due to the dummy noise. It would be interesting to come up with a faster and a more consistent protocol that can deal with issue.

An example of this setting can be seen in hospital records. Hospitals maintain electronic health records (EHRs) that contain sensitive patient information, including medical history, diagnoses, treatments, and billing information. Due to the Health Insurance Portability and Accountability Act (HIPAA), access to these records is strictly controlled and typically limited to authorized personnel to protect patient privacy. Since an area can have many such hospitals, these hospitals act as sensitive decentralized silos of data and any institution (such as insurance agencies or bank loan providers) which requires information from such hospitals and since the respective institutions need such data to train their own models and the data that they posses act as the "public silo". We can also extend this application into settings where certain countries may not allow the sensitive data to leave the country (eg. E.U. GDPR) but may require a fair model for their decision driven processes.

