# OpenReview forum: "A Stochastic Optimization Framework for Private and Fair Learning From Decentralized Data"
_TMLR — Under review for TMLR_

### Review · Reviewer_qZng · 2026-04-21

**Summary Of Contributions:**

This paper proposes an algorithm for conducting fair and differentially private federated learning. The core idea of the paper is that each client (silo) has a set of public data points $\{x_i, y_i\}$ and a corresponding set of private labels $s_i$. The goal is to learn a predictor $\hat y_\theta: \mathcal X \to \mathcal Y$ such that the predictions are statistically independent of the sensitive attributes. The authors present an algorithm for learning this predictor while also regularizing to enforce a fairness constraint. They demonstrate the utility of their approach on real datasets.

**Audience:**

Yes

**Audience Explanation:**

This paper addresses the intersection between federated learning, fairness, and privacy, which I feel would interest a sub-community of the TMLR audience.

**Claims And Evidence:**

No

**Claims Explanation:**

Overall, I think the paper tackles an interesting problem, and it seems to be taking a reasonable approach. However, I had a number of questions regarding the writing, that prevented me from being able to confidently state that the work is technically correct. I list these below.

1) The privacy model seems to be a bit stylized. In practice, I would expect that (X,Y) could be correlated with S, but the privacy metric defines neighboring datasets as those that vary only in terms of $s_i \neq s_i’$. It is not clear to me how releasing information about (x,y) without noise is not implicitly leaking information about s. (i.e., Algorithm 1 is not adding noise to the $g$ gradients that depends on their sensitivity, because sensitivity is defined only with respect to $s$) For example, what if X and S have a one-to-one relationship in practice. Then leaking information about X is also leaking information about S. It seems like the paper’s setting could be well-suited to an inferential privacy framework.

2) Setting aside questions of the privacy definition and formulation, my main question about this work is that I don't understand what fairness guarantee we are actually getting. The predictor loss is regularized with a $\chi^2$ term that somehow quantifies "how unfair" the predictor is, but there is no final theoretical result on the degree of unfairness provided by the final predictor---only privacy and gradient norm (with respect to $\theta$). Am I understanding this correctly? If so, I think this should be emphasized and explained explicitly in the paper.

3) There were many places where I did not understand the notation. First, the description of the fairness metric was confusing---can you please write out the precise definition of your fairness metric? Right now it is written in English, so it is difficult to understand exactly what you mean. In particular, it is currently written in terms of a predictor $\mathcal A:\mathcal Z \to \mathcal Y$. However by definition, $Z=(X,Y)$ (beginning of Sec. 2), so it's not clear to me what $\mathcal A$ is predicting here. (Note that later, the paper defines Z=(X,Y,S) at the bottom of page 3. Which is it?) Did you mean $\mathcal A: \mathcal X \to \mathcal Y$? Even without this point of confusion, I think it would be better to precisely write out your fairness definition with notation.

Another notational confusion: when you present the regularizer $\cal D$, it wasn’t clear at first that you are using the function $D_R$ as an instantiation of $\cal D$ because the notation is different.

I could not find a precise definition of $\Pi_{\mathcal W}$ in the writeup (Algorithm 1). Can you please clarify what this function is? I assume it is a projection onto $\mathcal W$, but I don’t understand how exactly $\mathcal W$ is defined. In the preamble to Algorithm 1, it says “set $\mathcal W \subset \mathbb R^{k\times l}$”. What does this mean? Don’t you need $|W_{r,q}| \leq D$ for Theorem 6 to hold?

Can you please explain how the global sensitivity of your gradients with respect to $W$ and $\theta$ are bounded? The Appendix provides a result from prior work stating this to be true, but it’s not clear to me from the algorithm where the sensitivity is controlled. For W, I think it may be happening with the $\Pi_{\mathcal W}$ function, but I’m not sure since I can’t find a definition for the function. For $\theta$, I didn’t understand why the gradient is bounded, and hence how you can use the Gaussian mechanism.

There were many details that were not fully explained in the manuscript, often referring to prior work. This makes the paper feel like it is not self-contained. The global sensitivity bounds above are one example. Another is that the authors talk about a centralized algorithm from (Lowy et al, 2023) that cannot be used because it would "leak central data to the server". What does the algorithm do that is different from the one proposed in that work? This seems like a closely-related prior work, but the algorithmic differences are not clearly stated.

**Requested Changes:**

Please take a pass at the following:

1) Update/clarify the fairness definition in your formulation.

2) Please revise the notation to make sure that all terms are clearly-defined and consistent with one another. See review for an incomplete list of confusing notation.

3) Justify the privacy model more thoroughly.

4) Explain how the algorithm ensures that gradients with respect to $\theta$ and $W$ are bounded, giving rise to the added noise in Algorithm 1.

---

> ### Author Response · Authors · 2026-07-11
>
> We would like to thank the reviewer for their feedback on the paper. Please find the responses to your concerns attached below.
>
> >The privacy model seems to be a bit...
>
> We agree that the privacy guarantee in our paper should be interpreted carefully. Our formal privacy notion is differential privacy with respect to the sensitive attributes S, rather than full-record differential privacy for (X,Y,S). In particular, our adjacency relation fixes (X,Y) and changes only the sensitive attribute S. Therefore, the guarantee should be read as protecting the additional information about S that is revealed through the use of the sensitive attributes in the fairness regularizer, conditional on the non-sensitive features and labels. This setting of privatizing only certain subset of attributes has been discussed in existing works like Ghazi et. Al.
>
> This also explains why Algorithm 1 does not add noise to the ordinary loss gradients g_{t,j}: these gradients depend only on (X,Y), and hence have zero sensitivity under our stated neighboring relation. The sensitive gradients involving the fairness regularizer do depend on S, so these are the gradients to which Gaussian noise is added.
>
> We agree with the reviewer that if (X,Y) are strongly correlated with S, then an adversary may infer information about S from (X,Y)-dependent quantities alone. In the extreme case where X determines S, releasing information about X can indeed reveal S. This is not ruled out by our privacy definition, and our current guarantee is not meant to provide inferential privacy against such correlations. Rather, it is a sensitive-attribute DP guarantee, following the privacy model used in prior DP fair learning work, where demographic attributes are treated as restricted/sensitive while the remaining features and labels are assumed to be available for training.
>
> We have revised the paper to make this distinction explicit after Definition 3 on Page 4 of our paper.
>
> [1] Ghazi, B., Golowich, N., Kumar, R., Manurangsi, P., & Zhang, C. (2021). Deep learning with label differential privacy. Advances in neural information processing systems, 34, 27131-27145.
>
> > Setting aside questions of the privacy definition and formulation, my main ...
>
> We agree that the current presentation should more clearly distinguish between the role of the fairness regularizer and the formal guarantee proved in the paper. Our algorithm promotes fairness by optimizing a fairness-regularized objective, where the regularizer is a chi-squared-dependence surrogate for demographic parity or equalized odds. This surrogate is used because prior work shows that small dependence between the predictions and the sensitive attributes implies small violations of the corresponding group-fairness notion. Thus, the role of the regularizer is to encourage the learned predictor to reduce demographic parity or equalized odds violations.
>
> However, we agree that our main theorem, as currently stated, is not a direct final-predictor fairness guarantee. Rather, it proves that the proposed algorithm satisfies ISRL-DP and converges to an approximate stationary point of the fairness-regularized objective. In particular, the theorem does not by itself state that the returned predictor has a bounded demographic parity or equalized odds violation. The observed fairness improvements are demonstrated empirically through the fairness-accuracy tradeoff curves. We have added a paragraph in the paper to make this distinction explicit after Theorem 7.
>
> > There were many places where I did not understand the notation. First, th
>
> The definition of the fairness violation is given in Equation (2) on page 3 for demographic parity, while the binary equalized odds violation is defined in Appendix E.1. The corresponding $\chi^2$-based fairness regularizers for both notions are provided in Appendix A.1. We have clarified these references in the main paper to make them easier to locate.
>
> Regarding the predictor, you are correct that it is a mapping from the input space $\mathcal{X}$ to the label space $\mathcal{Y}$. We have corrected the notation throughout the paper to reflect this.
>
> > Another notational confusion: when you present the...
>
> We have made the notation consistent in the revised version of the paper.

---

> > ### Author Response · Authors · 2026-07-11
> >
> > > I could not find a precise...
> > > Can you please explain how the global sensitivity of your gradients with respect to...
> >
> > Since these questions are closely related, we address them together.
> >
> > First, the reviewer is correct that $\Pi_{\mathcal{W}}$ denotes projection onto the constraint set $\mathcal W$. We have revised the paper to define this explicitly as the Euclidean projection, with respect to the Frobenius norm.
> >
> > We also agree that the boundedness condition on $W$ should be stated more clearly. In Theorem 6, the privacy analysis requires that the iterates satisfy |W_t_{r,q}| <= D for all coordinates $(r,q)$ and all iterations $t$ and the fact that the outer function is $L_\theta$ lipschitz in order for the sensitivity guarantees to hold. We have updated the statement in the Lemma 10 to include the lipschitzness of $\mathcal{F}$ and coordinate wise upper bound of $W$ so that the connection is more clear and there is no confusion. We have also rewritten the coordinate-wise bound for $W$ to be $R$ so that there is distinction between the diameter of the constraint set and the per-coordinate bound.
> >
> > Theoretically, we assume that the derivatives with respect to \(\theta\) are uniformly bounded through the $L_\theta$-Lipschitz assumption in Assumption 9(1). Such Lipschitz assumptions are standard in the analysis of differentially private stochastic optimization and DP-SGD [1]. Combined with the per-coordinate boundedness of $W$, this is precisely the setting considered by Lowy et al. (2023), which establishes the required global sensitivity bound for the gradient of the fairness regularizer with respect to $\theta$. It is important to note that we use these sensitivity calculations for each silo.
> >
> > In practice, bounded per-sample updates are enforced via per-sample gradient clipping, following the standard implementation of DP-SGD [2]. This ensures that the sensitivity of the released gradients is bounded before computing the gradient of the regularizer. We have updated this step in the algorithm by sending clipped per-sample gradients to the sensitive silo. For the $W$-updates, our utility analysis requires the projection set $\mathcal W$ to have bounded diameter, while the privacy analysis requires each coordinate of $W$ to be bounded by $D$. To satisfy both requirements simultaneously, we choose $\mathcal W$ to be the Euclidean ball of radius $D$ centered at the origin and projected onto this set after every update. Since $diam(\mathcal W)=2D$, this only changes the constant appearing in the utility bound, while preserving the stated asymptotic guarantee. We have added the gradients of the regularizer with respect to the parameters for clarity in Appendix B.
> >
> > [1] Bassily, R., Smith, A., & Thakurta, A. (2014, October). Private empirical risk minimization: Efficient algorithms and tight error bounds. In 2014 IEEE 55th annual symposium on foundations of computer science (pp. 464-473). IEEE.
> > [2] Abadi, M., Chu, A., Goodfellow, I., McMahan, H. B., Mironov, I., Talwar, K., & Zhang, L. (2016, October). Deep learning with differential privacy. In Proceedings of the 2016 ACM SIGSAC conference on computer and communications security (pp. 308-318).
> >
> > > There were many details that were not fully explained in the manuscri..
> >
> > We agree that some technical details were stated too briefly and that the relationship to Lowy et al. (2023) should be made more explicit. We have revised the manuscript to make the paper more self-contained, including the definitions of the projection operator, the role of the constraint set \(\mathcal W\), and the sensitivity bounds used in the privacy analysis.
> > Our work builds on the chi-squared fairness regularizer and min--max reformulation of Lowy et al. (2023). We do not intend to claim these tools as new. Rather, our contribution is to show how these tools can be adapted to federated and hybrid-centralization settings while satisfying ISRL-DP. The centralized algorithm of Lowy et al. (2023) assumes access to the relevant data in a trusted central location. If applied directly in our federated setting, it would require either centralizing the sensitive data or send sensitive-gradient information to the server, which is not compatible with the ISRL-DP requirement that each silo's outgoing transcript be private. Thus, to accommodate for this change Algorithm 1 differs in some aspects: each silo computes the sensitive-gradient terms locally, adds Gaussian noise before communication, and the server only aggregates the noisy sensitive gradients together with the non-sensitive loss gradients. The analysis must therefore account for distributed mini-batch sampling across silos, independent noise added at each silo, and the stronger transcript-level ISRL-DP guarantee. We have added this discussion on Page 7.

---

### Review · Reviewer_9rJo · 2026-05-22

**Summary Of Contributions:**

Cross-Silo Federated Learning (DP) can enable entities with sensitive data to collaboratively learn a model without sharing individual records in the clear. Differentially private (DP) cross-silo FL can allow this process to provably protect the privacy of individual records on which this model is trained. This is the setup of the current work. Each of N silos has a number of user records. The silos interact to collaboratively train a model and the goal is to ensure that all the communication out of a silo is DP with respect to the records stored in this silo. The current work is concerned with fairness in this setting.

The main result in the work is to show an algorithm that can satisfy record level DP in this setting, and can optimize an objective that is a sum of a training loss and a fairness term. The approach in this work is to define fairness as some kind of divergence measure (e.g. chi-squared) between the joint distribution (model output, sensitive attribute) and the product of the marginals. This is expected to help ensure demographic parity.

The fairness term in the objective can be phrased as a max of simpler terms, building on previous work of Lowy et al., and thus the whole objective can be phrased as a min-max objective. This allows the optimization to use stochastic gradient based updates, which are easy to adapt to the federated setting. DP is ensured by adding Gaussian noise on each silo’s contribution. The authors show that this algorithm with appropriate parameters satisfied (eps,delta)-DP and provably converges to an approximate stationary point of the regularized objective function. They show empirically that the algorithm works reasonably well compared to some previous algorithms for this task.

**Audience:**

Yes

**Audience Explanation:**

I can imagine some readers finding it useful that a gradient-based fair DP learning method can be adapted to the federated setting in the natural way.

**Broader Impact Concerns:**

No concerns

**Claims And Evidence:**

No

**Claims Explanation:**

Strengths:
- Clear problem definition, and formal algorithms.
Weaknesses:
- The paper builds on previous work (Lowy et al. 2023), which proposed an algorithm for DP fair learning. To my understanding, the only change in this work is to extend this to the federated setting by replacing the noise generated centrally for DP with each silo generating the same noise. This is equivalent to increasing the noise variance by a \sqrt{N} factor and thus the error bound in Thm 7 is a factor \sqrt{N} worse than the corresponding theorem in Lowy et al. This is too incremental a contribution to be worth publishing. The paper claims that they "develop a novel framework" and I find this claim to be vastly overstated.
- In light of this, I find the experiments somewhat puzzling. The authors claim that “our algorithm even outperforms strong centralized DP fair baselines”. This claim does not seem valid to me. The experimental part of the paper does not compare with Lowy et al., and it seems clear to me that their algorithm (being identical to Lowy et al. except adding more noise) will be worse than Lowy et al.

In light of the weaknesses, I recommend rejection.

**Requested Changes:**

See weaknesses above

---

> ### Author Response · Authors · 2026-07-11
>
> We would like to thank the reviewer for their feedback on the paper. Please find the responses to your concerns attached below.
>
> > The paper builds on previous work (Lowy et al. 2023), which proposed an algorithm for DP fair learning. To my understanding, the only change in this work is to extend this to the federated setting by replacing the noise generated centrally for DP with each silo generating the same noise. This is equivalent to increasing the noise variance by a \sqrt{N} factor and thus the error bound in Thm 7 is a factor \sqrt{N} worse than the corresponding theorem in Lowy et al. This is too incremental a contribution to be worth publishing. The paper claims that they "develop a novel framework" and I find this claim to be vastly overstated.
>
> We agree that the wording in the paper should be revised to avoid suggesting that we introduce a fundamentally new centralized DP-fair learning algorithm independent of Lowy et al. The main contribution of our work is an algorithmic framework that simultaneously addresses differential privacy, fairness, and federated learning in a siloed setting, where data cannot be pooled centrally. To the best of our knowledge, this combination has not been previously studied in the DP-fair learning literature. Thus, the value of the work is not in providing a stronger centralized DP-fair algorithm, but in showing that DP-fair learning can be realized in the federated setting with provable privacy, fairness, and utility guarantees, while also quantifying the precise utility cost incurred by decentralization. We have revised the paper to make this positioning clearer, explicitly acknowledge the role of Lowy et al. and replace phrases such as “develop a novel framework” with a more precise statement emphasizing that we provide an algorithm for DP-fair federated learning.
>
> > In light of this, I find the experiments somewhat puzzling. The authors claim that “our algorithm even outperforms strong centralized DP fair baselines”. This claim does not seem valid to me. The experimental part of the paper does not compare with Lowy et al., and it seems clear to me that their algorithm (being identical to Lowy et al. except adding more noise) will be worse than Lowy et al.
>
> We agree with the reviewer that the statement, as written, is too strong. Our intention was not to claim that our method uniformly outperforms the strongest centralized DP-fair learning baselines, nor to imply a direct comparison with Lowy et al. Since our experimental section does not include Lowy et al., the current wording does not support such a claim.
> We will revise the paper accordingly. The intended message was that, compared to a substantial subset of existing centralized DP-fair frameworks considered in our experiments, our federated framework achieves better utility while operating in the federated setting. We also note that there is relatively limited prior work on DP-fair learning in the federated setting, which restricts the set of directly comparable baselines. We will therefore remove the claim that our algorithm outperforms existing strong centralized DP-fair baselines and replace it with a more precise statement emphasizing that our method is competitive with, and in some cases improves upon, some significant centralized DP-fair baselines included in our evaluation.

---

### Review · Reviewer_ZGVZ · 2026-06-26

**Summary Of Contributions:**

This paper studies fair federated learning under so-called inter-silo record-level differential privacy (ISRL-DP). The authors combine an existing stochastic fairness optimization framework (based on a min-max reformulation of a fairness regularizer) with a federated optimization protocol in which each silo perturbs its locally-computed sensitive gradients to satisfy ISRL-DP. The paper establishes convergence guarantees for smooth (possibly nonconvex) objectives using stochastic minibatches and demonstrates empirical improvements in fairness-accuracy tradeoffs over previous fair federated learning baselines.

From a technical perspective, much of the optimization framework (the fairness objective, the min-max reformulation, and the stochastic optimization methodology) is inherited from the authors’ prior work on centralized fair learning. The primary contribution of this paper is extending that framework to the cross-silo federated setting under ISRL-DP and providing the corresponding convergence analysis. The paper also derives a convergence result for ISRL-DP optimization of nonconvex-strongly-concave min-max problems, which may be of independent interest beyond the fairness application.

## Strengths:
- Addresses a practical problem at the intersection of federated learning, fairness, and differential privacy.
- Provides a complete algorithm together with privacy and theoretical convergence guarantees.
- The optimization theorem may be useful beyond the specific fairness application.
- Empirical results demonstrate improved fairness-accuracy tradeoffs.
- The authors discuss how the framework extends beyond the standard horizontal federated setting to accommodate several different partitions of sensitive and non-sensitive data, including hybrid centralized/decentralized settings, which I think makes it a more interesting framework conceptually.

## Weaknesses:
- The core optimization technique is built upon the authors’ previous work in the centralized setting, so the novelty primarily lies in the federated extension and its analysis.
- The privacy model protects only the sensitive attribute while treating the remaining features and labels as public. Since the non-sensitive features may themselves be highly correlated with the sensitive attribute, exactly the setting in which fairness is most relevant, it would be helpful for the paper to discuss the practical implications and limitations of this modeling choice.

**Audience:**

Yes

**Audience Explanation:**

I expect researchers working on privacy-preserving machine learning, federated learning, or algorithmic fairness to find the results useful.

**Broader Impact Concerns:**

I do not have major broader-impact concern. The authors may wish to stress that the privacy guarantees apply only to the designated sensitive attributes and do not prevent inference of those attributes from correlated non-sensitive features.

**Claims And Evidence:**

Yes

**Claims Explanation:**

The paper provides formal privacy guarantees, convergence analysis, and experimental evaluations supporting its empirical claims. The theoretical arguments seem correct, if we trust the components that come from the FERMI-based work.

**Requested Changes:**

Important:
- Better explain the trust model and the ISRL-DP privacy definition, perhaps with an example. In particular the phrase "Thus, adjacent distributed datasets Z and Z1 may di ffer in up to N samples, one from each silo." may be confusing because we are not protecting against N changes simultaneously (as if Alice was part of all N silos), but a single change in a single silo's dataset, with all other datasets fixed.
- Discuss a bit more the privacy and fairness model (mis)match. The paper protects only the sensitive attribute while treating the features and labels as fixed. This is an important modeling assumption that should be discussed more explicitly, including its practical implications and limitations related to fairness.

Suggested improvements:
- Consider replacing |B_t| with m in the theorems.
- More clearly distinguish the novel contributions from prior work, especially relative to the authors’ previous FERMI and centralized DP fairness papers.
- Perhaps include a theorem for when all features are sensitive.
- Perhaps add in Section 4 how the analysis extends to unequal silo datasets.

---

> ### Author Response · Authors · 2026-07-11
>
> > Better explain the trust model and the ISRL-DP privacy definition, perhaps with an example. In particular the phrase "Thus, adjacent distributed datasets Z and Z1 may differ in up to N samples, one from each silo." may be confusing because we are not protecting against N changes simultaneously (as if Alice was part of all N silos), but a single change in a single silo's dataset, with all other datasets fixed.
>
> We agree that the current wording around adjacency in Definition 3 is incorrect and has been removed. Our intended privacy notion is a per-silo record-level guarantee: for each target silo $j$, the transcript sent by silo $j$ should be differentially private with respect to a change in one individual's sensitive attribute in that silo, while all other silos' datasets are fixed. Thus, we are not modeling a single individual as appearing in all $N$ silos, nor are we claiming protection against $N$ simultaneous record changes without the usual composition or group-privacy degradation. We have added a note after the definition to make this explicit.
>
> > Discuss a bit more the privacy and fairness model (mis)match. The paper protects only the sensitive attribute while treating the features and labels as fixed. This is an important modeling assumption that should be discussed more explicitly, including its practical implications and limitations related to fairness.
>
> Practically, DP with respect to sensitive data allows the algorithm to use protected attributes, such as race, gender, or age, for fairness regularization while limiting how much the training transcript can reveal about any one individual's sensitive attribute. This is useful in settings where the features and labels are available for model training, but demographic attributes are restricted or legally sensitive. However, this guarantee does not protect the non-sensitive features or labels themselves, nor does it prevent inference of the sensitive attribute from strong proxies in those features. We have added this discussion on Page 4 about the practical implications of our approach and limitations of our modelling assumption and how it doesn't necessarily protect against proxy based inference. Section 4 highlights some ways on how our framework could be extended into different settings and its practical usage.
>
>
> > Consider replacing |B_t| with m in the theorems.
>
> We have made these edits.
>
> > More clearly distinguish the novel contributions from prior work, especially relative to the authors’ previous FERMI and centralized DP fairness papers.
>
> We thank the reviewer for this suggestion. Our work builds on the FERMI framework in the sense that we use the FERMI objective as the underlying fairness-aware learning objective. However, FERMI itself does not address differential privacy or federated/siloed learning. The main contribution of the present work is to provide a privacy-preserving optimization procedure for the FERMI objective in the federated setting, where data are distributed across silos and cannot be pooled centrally. In particular, we show how to optimize this fairness objective while simultaneously ensuring differential privacy and maintaining provable fairness and utility guarantees.
>
> Regarding the comparison with Lowy et al. (2023), we have expanded the discussion comparing our work with Lowy et al. (2023) on Page 7. Lowy et al. study centralized DP fair learning in a trusted-curator setting, whereas our work considers the federated/siloed setting and quantifies the additional utility cost incurred by decentralization and distributed noise generation.
>
> > Perhaps include a theorem for when all features are sensitive.
>
> We have added a theorem in the Appendix C.1 which extends the algorithm for the case when all features are sensitive. It is important to note that one can also directly get DP guarantees for all features private if we consider the loss to be the regularized loss in total and per-sample clipping is done with respect to both $\theta$ and W separately. The privacy and utility analysis would follow directly from (now) Theorem 14.
>
> > Perhaps add in Section 4 how the analysis extends to unequal silo datasets.
>
> We have provided a sketch in Section 4 on how the analysis can be extended to unequal silo datasets.